# The structural determinants of PH domain-mediated regulation of Akt revealed by segmental labeling

Nam Chu[1,2,3†], Thibault Viennet[2,4†], Hwan Bae[1,2], Antonieta Salguero[1,2,3], Andras Boeszoermenyi[2,4], Haribabu Arthanari[2,4*], Philip A Cole[1,2,3*]

[1]Division of Genetics, Department of Medicine, Brigham and Women's Hospital, Boston, United States; [2]Department of Biological Chemistry and Molecular Pharmacology, Harvard Medical School, Boston, United States; [3]Department of Pharmacology and Molecular Sciences, Johns Hopkins School of Medicine, Baltimore, United States; [4]Department of Cancer Biology, Dana-Farber Cancer Institute, Boston, United States

**Abstract** Akt is a critical protein kinase that governs cancer cell growth and metabolism. Akt appears to be autoinhibited by an intramolecular interaction between its N-terminal pleckstrin homology (PH) domain and kinase domain, which is relieved by C-tail phosphorylation, but the precise molecular mechanisms remain elusive. Here, we use a combination of protein semisynthesis, NMR, and enzymological analysis to characterize structural features of the PH domain in its autoinhibited and activated states. We find that Akt autoinhibition depends on the length/flexibility of the PH-kinase linker. We identify a role for a dynamic short segment in the PH domain that appears to regulate autoinhibition and PDK1-catalyzed phosphorylation of Thr308 in the activation loop. We determine that Akt allosteric inhibitor MK2206 drives distinct PH domain structural changes compared to baseline autoinhibited Akt. These results highlight how the conformational plasticity of Akt governs the delicate control of its catalytic properties.

**\*For correspondence:**
hari_arthanari@hms.harvard.edu (HA);
pacole@bwh.harvard.edu (PAC)

†These authors contributed equally to this work

## Introduction

Akt1 (termed Akt in the present work) is a Ser/Thr kinase that is a critical node in cell signaling and connects growth factor receptor activation to cell growth and metabolic regulation (*Manning and Toker, 2017*; *Liao and Hung, 2010*; *Fruman et al., 2017*). The Akt subfamily of kinases includes closely related paralogs Akt1-3 and are members of the larger AGC kinase family, comprised of about 60 members of the kinome (*Leroux et al., 2018*; *Pearce et al., 2010*). Akt is a 480 amino acid protein that contains an N-terminal PH (pleckstrin homology) domain followed by a catalytic (kinase) domain and culminates in a regulatory C-terminal disordered tail (C-tail) (*Figure 1A*). Our current understanding of Akt activation is that, in response to growth factor stimulation, phosphatidyl inositol diphosphate (PIP2) is converted to PIP3 by PI3-kinase and PIP3 recruits Akt to the plasma membrane via its PH domain (*Manning and Toker, 2017*; *Haeusler et al., 2018*). Upon recruitment, Akt is phosphorylated by both mTORC2 kinase at Ser473 and PDK1 kinase at Thr308 and this dual phosphorylation leads to activation of Akt (*Sarbassov, 2005*; *Alessi et al., 1996*). Additionally, Akt can also be phosphorylated at Ser473 by DNA-PK upon activation of DNA damage response in the nucleus (*Bozulic et al., 2008*). In an alternative pathway of Akt activation, Akt may be phosphorylated on Ser477 and Thr479 by Cdk2/cyclinA instead of Ser473 and this may happen in the cell nucleus (*Liu et al., 2014*). Excluding Thr308, the three other phosphorylations (Ser473, Ser477 and Thr479) are located in the disordered C-terminal tail.

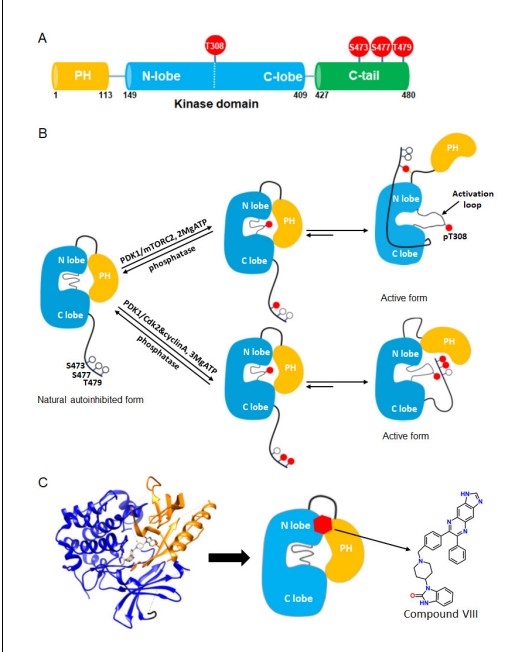

**Figure 1.** Akt domain architecture, two distinct activation mechanisms mediated by C-tail phosphorylations, and the current model for drug-induced autoinhibited form. (**A**) Schematic representation of Akt domains with phosphorylation sites of interest highlighted as red balls. (**B**) Cartoon model depicted for two distinct Akt activation mechanisms induced by phospho-Ser473 and dual phospho-Ser477/Thr479 (*Chu et al., 2018*). Without C-tail phosphorylations, Akt remains in an inactive autoinhibited state where the PH and kinase domains interact intramolecularly. The mTORC2-mediated Ser473 phosphorylation activates Akt by inducing an interaction between the C-tail and the PH-kinase domain linker, dislodging the PH domain from the kinase domain. Alternatively, the dual Cdk2/cyclinA-mediated pSer477/pThr479 is presumed to activate Akt by interaction with the activation loop and PH domain. Note that, although C-terminal phosphorylation is shown preceding activation loop Thr308 phosphorylation by PDK1, it is uncertain in normal cell signaling whether C-terminal phosphorylation or Thr308 phosphorylation comes first in Akt activation or if it is in random order. (**C**) Crystal structure of Akt (aa 1–443) bound to allosteric inhibitor compound VIII (left, PDB: 3O96, [*Wu et al., 2010*]) and cartoon model illustrating the current model of the allosteric drug-induced autoinhibited form of Akt with the non-phosphorylated C-tail.

Over the past two decades, several aspects of the regulatory mechanisms of Akt have been described. An autoinhibitory mechanism involving an intramolecular interaction between its N-terminal PH domain and C-terminal kinase domain is generally accepted (*Bellacosa et al., 1998*; *Calleja et al., 2009*; *Calleja et al., 2007*; *Cole et al., 2019*; *Figure 1B*). The structural basis for this autoinhibitory conformation has been proposed based on high resolution X-ray crystal structures of near full-length Akt in complex with small molecule catalytic inhibitors such as compound VIII (*Wu et al., 2010*; *Lapierre et al., 2016*; *Ashwell et al., 2012*; *Figure 1C*). A mechanistically related allosteric inhibitor of compound VIII, MK2206, has been widely used in cancer therapeutic clinical trials aimed to block Akt action (*Hirai et al., 2010*; *Sangai et al., 2012*; *Yap et al., 2011*; *Nitulescu et al., 2016*).

While phosphorylation at Thr308 is important for kinase activity in the context of the isolated Akt kinase domain, C-terminal phosphorylation shows significant impact only in the context of full-length Akt (*Chu et al., 2018*; *Cole et al., 2019*; *Alessi et al., 1996*; *Calleja et al., 2009*; *Yang et al., 2002a*; *Yang et al., 2002b*). Recent structural studies suggest that phosphorylation at Ser473 concurrently interacts with a PH-kinase domain linker basic patch, especially the side-chain of Arg144, and the kinase N-lobe via Gln218 (*Chu et al., 2018*; *Figure 1B*). A model based on these structural interactions postulates that these interactions with phosphorylated Ser473 (pSer473) help to displace the PH domain from the kinase domain thereby stimulating catalytic activity. Phosphorylation of Ser477/Thr479 is suggested to relieve autoinhibition of Akt in a distinct fashion, potentially by interaction with the activation loop and/or the PH domain (*Chu et al., 2018*; *Figure 1B*).

However, the roles and conformational status of the PH domain in the active and inactive states of Akt have been controversial. It has been suggested that PIP3 activates Akt not simply by membrane recruitment but by dislodging the PH domain from the kinase domain independent of promoting Akt phosphorylation by upstream kinases (*Ebner et al., 2017*; *Lučić et al., 2018*). In contrast to these studies, this allosteric mechanism of Akt activation by PIP3 was not confirmed by our group where neither soluble nor vesicle-embedded PIP3 stimulated Akt kinase activity (*Chu et al., 2018*). Moreover, the binding affinities of soluble PIP3 with pSer473 Akt vs. non-phospho C-tail Akt were essentially identical (*Chu et al., 2018*). In comparison, soluble PIP3 binding to pSer477/pThr479 was attenuated by about fourfold compared to pSer473 and non-C-terminally phosphorylated Akt (*Chu et al., 2018*). Remarkably, crystal structures of near full-length

Akt in complex with compound VIII and related allosteric inhibitors reveal that the PIP3 binding pocket on the PH domain is occluded by its interactions with the kinase domain (*Wu et al., 2010*; *Figure 1C*).

X-ray crystal structures of near full-length Akt in the absence of allosteric inhibitors, as well as of active Akt including the PH domain have not yet been reported. Thus, our understanding of the structure and conformation of the PH domain in the context of the full-length Akt protein is incomplete. In this study, we have used a combination of enzymatic analysis of linker-altered Akts and solution NMR analysis of segmentally isotopically labeled semisynthetic Akt forms to characterize the structural features of the PH domain within the full-length protein in differentially phosphorylated states. These structural measurements combined with biochemical data highlight regions in the PH domain that appear to show distinct changes in the context of C-terminal phosphorylation and allosteric drug inhibition.

## Results

### Role of linker length and flexibility in autoinhibition and activation of Akt

Intramolecular interactions between the Akt PH and kinase domains are presumed to enforce autoinhibition of Akt's enzymatic activity. Interaction of pSer473 with the linker basic patch, particularly Arg144, appears to help relieve such autoinhibition. We have attempted to determine the role of linker length and flexibility that may contribute to how the pSer473-Arg144 interaction influences the relief of autoinhibition. A prior cellular study meant to investigate this possibility by inserting a flexible hexa-Gly segment in the middle of the PH-kinase linker showed reduced Akt signaling in response to growth factors. The results of these experiments were complicated to interpret, however, because hexa-Gly insertion led to reduced pSer473 levels (*Chu et al., 2018*).

To characterize the role of linker flexibility in more depth, we prepared the semisynthetic wt and hexa-Gly insertion Akt forms containing C-terminal phosphorylation using expressed protein ligation (*Muir et al., 1998*; *Figure 2—figure supplement 1*). In this procedure, aa1-459 of wt Akt and the hexa-Gly insertion segments fused to an intein were expressed in insect cells along with PDK1 and phosphatase inhibitor okadaic acid to ensure complete Thr308 phosphorylation. The C-terminal thioester fragments were ligated to N-Cys C-tail peptides (aa460-480) containing pSer473, pSer477/pThr479, or no C-tail phosphorylation and the semisynthetic proteins further purified to greater than 80%. Near stoichiometric Thr308 phosphorylation of these Akt forms was confirmed by western blot. It should be noted that Thr450 phosphorylation which is necessary for Akt stability occurs in insect cells concurrently with Akt production (*Chu et al., 2018*).

We then analyzed the catalytic properties of the wild type (wt) and hexaGly mutant proteins (*Figure 2A*). It should be noted that the apparent wt catalytic efficiencies ($k_{cat}/K_m$ values) were relatively similar to those reported previously for wt Akt in various states of phosphorylation generated in this fashion (within ~3-fold, [*Chu et al., 2018*]). Some variation in kinase activities between different preparations can be explained by possible post-translational modification

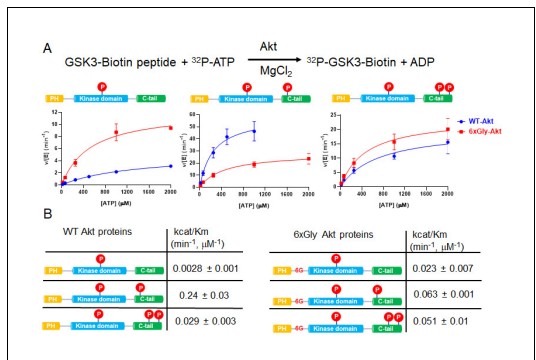

**Figure 2.** The PH-kinase domain linker length/flexibility affects Akt activation. (**A**) Schematic illustration for radiometric kinase assay using a biotinylated GSK3 peptide as Akt substrate and steady-state kinetic plots for v/[E] versus [ATP] with 20 µM GSK3 peptide for semisynthetic pThr308 Akt proteins WT (blue) versus the linker 6xGly insertion (red) with non-p C-tail (left), pSer473 (middle) and dual-pSer477/pThr479 (right), n = 2. (**B**) Enzyme catalytic efficiencies (apparent $k_{cat}/K_m$) obtained from (**A**) for each semisynthetic Akt protein, WT (left) and 6xGly insertion (right), two independent repeats were performed for each assay, S.D. shown.

The online version of this article includes the following figure supplement(s) for figure 2:

**Figure supplement 1.** Semisynthesis of hexa-Gly Akt proteins.

differences in the recombinant protein portion of the semisynthetic Akts which may occur in insect cells. To minimize the effects of catalytic differences among insect cell preparations, where possible we compared Akt forms with distinct C-terminal phosphorylation status prepared from the same batch of recombinant Akt thioester. Moreover, the small contamination (~10%) of slightly truncated Akt in these preps (*Figure 2—figure supplement 1*) is not expected to alter significantly the $k_{cat}/K_m$ values measured (see Materials and methods).

There was a 3.4-fold reduction in $k_{cat}/K_m$ of hexaGly pSer473 Akt relative to that of wt pSer473 Akt (*Figure 2B*). In contrast, the non-phosphorylated C-tail hexaGly Akt was ~8 fold more active compared with the corresponding wt Akt. These results support a role for linker length or flexibility as a contributing factor to activation of Akt orchestrated by pSer473-basic linker patch interaction. This also indicates that autoinhibition is alleviated in the non-C-tail phosphorylated state if this linker is too flexible.

In contrast to the results with pSer473, the activity of the dual C-terminal phosphorylated (pSer477/pThr479) Akt showed only a minor difference (<2 fold) when comparing the hexaGly linker insertion vs. wt Akt (*Figure 2B*). These findings suggest that linker length or flexibility is not an important factor in pSer477/pThr479-mediated Akt activation, consistent with an alternative mode of regulation for this dual phosphorylated C-tail form.

## Segmental isotopic labeling of Akt by protein semisynthesis

We embarked on an effort to further characterize the domain structural interactions between the PH domain and the kinase domain within Akt using solution NMR. However, given the size of full-length Akt (56 kDa), the poor expression yields and low activity of *E. coli*-expressed kinase domain and the difficulty of achieving site-specific phosphorylation, we chose to develop a new three-piece expressed protein ligation-based segmental labeling protocol (*Liu et al., 2009*; *Xu et al., 1999*; *Figure 3A* and see Materials and methods).

The PH domain (aa 1–121) is expressed in *E. coli* and isotopically labeled with ($^{13}C$), $^{15}N$ and $^{2}H$ to ensure optimal relaxation properties (*Figure 3—figure supplement 1A*). The linker-kinase domain segment (aa 122–459) was expressed in *Sf9* insect cells using the baculovirus system and phosphorylated at Thr308 in vitro using recombinant GST-PDK1, achieving site-selective phosphorylation and baseline activity. The kinase domain was not isotopically labeled, thus greatly reducing spectral crowding in context of the full-length Akt. The C-terminal tails (aa 460–480) were prepared by solid phase peptide synthesis in order to efficiently generate the desired phospho-forms of Akts: pSer473, pSer477/pThr479 or the non-phosphorylated C-tail variants (*Figure 2—figure supplement 1A–C*).

To obtain segmental labeling of full-length Akt, both the PH and kinase domains were fused to GyrA inteins, allowing C-terminal thioester

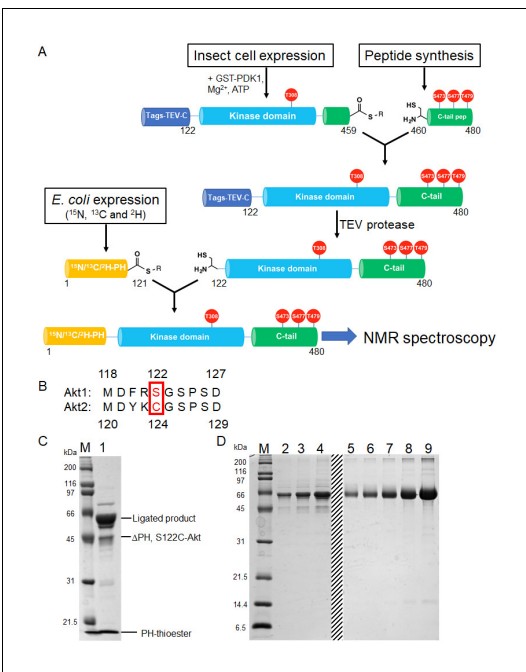

**Figure 3.** Semisynthesis strategy to generate segmentally $^{15}N$, $^{13}C$, $^{2}H$ isotopically labeled Akt containing distinct C-tail phospho forms. (**A**) Schematic representation of the semisynthesis strategy. (**B**) Alignment of Akt1 (aa 118–127) and Akt2 (aa 119–129) linkers with Akt1 Ser122 (ligation site, mutated to Cys in our study) and Akt2 Cys124 highlighted (red). (**C–D**) SDS-PAGE analyses of EPL reaction between S122C, Δ121, pThr308, pSer473 Akt fragment and isotopically labeled PH thioester fragment (**C**); and segmentally isotopically labeled pThr308, pSer473 full-length Akt purified from (**C**) using size exclusion chromatography (**D**), lanes 2–4: purified full-length Akt diluted 10-fold from stock, loaded volumes are 2.5, 5 and 10 µl, respectively, lanes 5–9: BSA standards 0.25, 0.5, 1, 2, 4 µg, dashed line: deletion of unrelated samples. M: protein markers (kDa).

The online version of this article includes the following figure supplement(s) for figure 3:

**Figure supplement 1.** Generation of segmentally $^{15}N$, $^{13}C$, $^{2}H$ isotopically labeled Akt proteins.

formation. The C-terminal thioester form of the Akt kinase domain was then reacted with the requisite C-terminal peptides possessing an N-terminal Cys residue. Then, a N-Cys on the kinase domain was exposed after TEV protease cleavage, and reacted with the PH domain C-terminal thioester. Note that the natural Ser122 of Akt1 was replaced by Cys at this position – as needed for chemoselective ligation during the assembly of the semisynthetic protein – but the corresponding native residue in Akt2 is a Cys, indicating that this substitution should be well-tolerated (*Figure 3b*). This sequential three-piece ligation strategy facilitated the production of milligram quantities of highly pure (monomeric), full-length Akt, which was triply labeled in the PH domain, with phosphorylation at Thr308 and specific and stoichiometric C-terminal phosphorylation states (*Figure 3C and D*, and *Figure 3—figure supplement 1B and C*). Kinase assays with segmentally labeled pSer473 Akt revealed that its enzymatic activity was within twofold of that of the unlabeled semisynthetic enzyme prepared in a two-piece ligation of aa1-459 with aa460-480 as described previously (*Chu et al., 2018*; *Figure 3—figure supplement 1D*).

## NMR analysis of segmentally labeled Akt forms

Given the limited solubility and stability of these full-length Akt forms under suitable NMR conditions, assignments were obtained for the isolated PH domain (see Materials and methods) and were subsequently transferred to the corresponding peaks in spectra of the segmentally labeled Akt proteins. About 60% of backbone hydrogen and nitrogen chemical shifts of residues in the segmentally labeled Akt forms could be unambiguously assigned using this procedure.

The interaction of the PH domain with the various phospho variants of the C-tail, in the context of the segmentally labeled full-length Akt, was initially assessed by $^{15}$N-$^{1}$H-HQSC spectra using combined chemical shift perturbations (CSP) and peak broadening (*Figure 4* and *Figure 5*). It is clear from a comparison of spectra of the isolated PH domain with that of the PH domain the context of full-length Akt (*Figure 5—figure supplement 1*) that the PH domain undergoes extensive interactions with the kinase domain and/or the C-tail, and potential conformational rearrangements. The CSP and line broadening are spread throughout the PH domain, including regions that interact with the PIP3 analog inositol hexaphosphate (IP6) (*Figure 5—figure supplement 2B and E*). Note that the CSPs observed on the PH domain are on average three-fold higher in the context of full-length Akt than those induced on the free PH domain by IP6 or the addition of the kinase domain in trans (*Figure 5—figure supplement 1* and *Figure 5—figure supplement 2A and D*). These findings highlight the importance of the intramolecular PH-kinase domain interactions in intact Akt that can be captured only in the context of the full-length protein.

Next we focused on comparing the spectral differences in the PH domains among the three C-tail phospho-states: pSer473, pSer477/pThr479, and non-phosphorylated C-tail (non-p-C) (*Figure 4* and *Figure 5*). While the N-terminus (aa 1–41, β-strands 1 to 3) seems to show relatively minor differences in response to variations in the C-tail phospho-states, the region encompassing β-strands 4 and 5 (aa 72–90) show comparatively larger perturbations depending on which C-tail phospho-states are present. The effects of CSP and line broadening can be induced by a direct interaction or an allosteric change, both of which result in changes in the electronic environment of the observed proton. To distinguish between direct and indirect binding interactions between the kinase and PH domains among the different Akt forms, we employed cross saturation transfer (CST) NMR measurements, which exclusively provides information on the direct interaction interface (*Wüthrich, 2000*; *Takahashi et al., 2000*). Perdeuterated, $^{15}$N-labeled, PH domain was expressed and covalently linked to the protonated kinase domain (and the various phospho-variants) as discussed above. In a CST experiment, we selectively magnetize the protonated protein in the aliphatic region, in this case the kinase domain, and monitor the transfer of magnetization to the perdeuterated partner, the PH domain. Since CST experiments are performed in 70% $^{2}$H$_2$O (30% H$_2$O) solvent to minimize non-specific effects due to spin diffusion, the inherent sensitivity of the experiment, which is concentration limited, is further reduced. Hence, we considered the CST effect in a contiguous stretch of amino acids as a true positive. Upon saturation of the kinase domain, peak intensities of select resonances in the PH domain were effectively reduced by as much as 90% (*Figure 5G* and *Figure 5—figure supplement 3*), underscoring the direct intramolecular interaction between the PH and kinase domains among the various Akt phospho-forms. The results from CST the experiments along with those derived from the CSP analysis are discussed below.

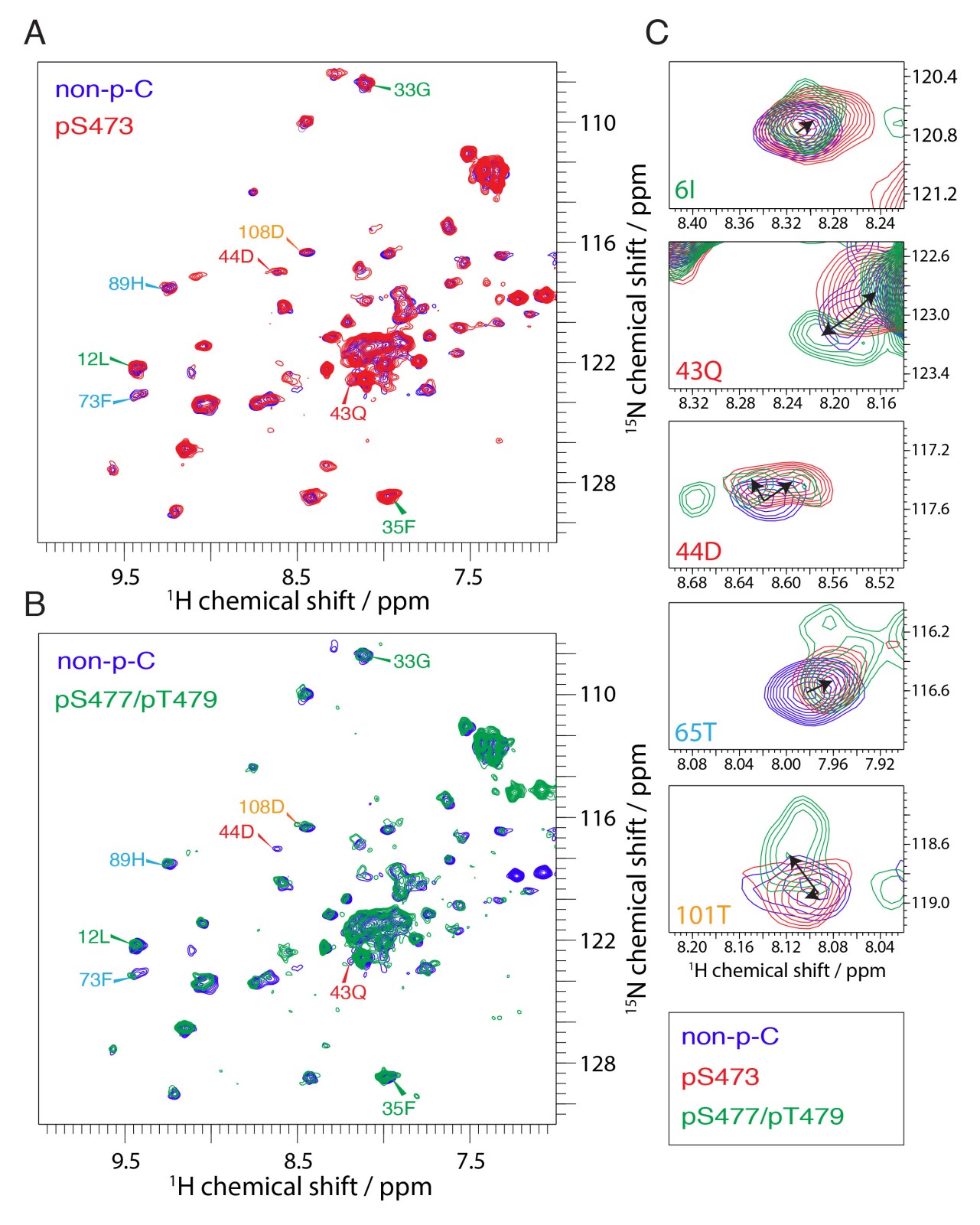

**Figure 4.** NMR reveals differences in the PH domain dependent on C-tail phospho states. (A) Overlay of $^{15}$N-$^{1}$H HSQC spectra of the PH domain in the context of full-length semisynthetic Akt with non-phosphorylated C-tail (blue) and pSer473 (red). Select residue-specific assignments are shown. (B) Overlay of $^{15}$N-$^{1}$H HSQC spectra of the PH domain in the context of full-length semisynthetic Akt with non-phosphorylated C-tail (blue) and pSer477/pThr479 (green). Select residue-specific assignments are shown. (C) Expanded spectra around the peaks assigned to Ile6, Gln43, Asp44, Thr65 and
*Figure 4 continued on next page*

*Figure 4 continued*
Thr101. Overlay of all three spectra, same color coding. Arrows indicate chemical shift perturbations. Note that contour levels have been plotted lower for the non-p-C Gln43 and pSer477/pThr479 Asp44 peaks.

Aligning with the CSP data, no striking differences, in a contiguous stretch of amino acids were observed in the regions encompassing β-strands 1–3 and 4–7 (*Figure 5G* and *Figure 5—figure supplement 3*). This indicates that the modes of activation of Akt may depend on subtle changes in the binding interface rather than on major differences in binding affinity. However, a small but distinct difference in the CST was observed for β-strands 4–7 across the three tested forms of Akt (*Figure 5G*). Specifically, the CST for β-strands 4–7 of pSer473 is lower than that of the non-phosphorylated and pSer477/pThr479 forms. This difference could be the result of a marginally weaker interaction of β-strands 4–7 of the PH domain with the kinase domain in the pSer473 form. This is consistent with the biochemical data.

Focusing on the strong CST effects, the C-terminal alpha-helix (aa 100–109) of the PH domain showed a significant CSP between the pSer473 and pSer477/pThr479 forms (*Figure 4* and *Figure 5B–E*). There were no significant changes in the chemical shifts of the C-terminal alpha-helix PH domain between the non-p-C and pSer473. On the contrary, the spectrum of pS477/pThr479 shows marked CSPs compared to the non-p-C form on the residues corresponding to the C-terminal helix. Thus, we surmise that this PH domain C-terminal alpha-helix plays a specific role in the mode of activation of Akt by the dual pSer477/pThr479 modifications. Notably, results from the CST experiments reinforced this possibility since the peaks of residues corresponding to the C-terminal helix exhibit marked reduction in intensity (near-complete disappearance) in the pSer477/pThr479 form (*Figure 5G* and *Figure 5—figure supplement 3*).

Furthermore, residues Gln43 and Asp44 of the PH domain which are in an extended loop between β-strands 3 and 4 show either relatively large CSPs or marked peak broadening beyond detection in the phospho-C-tail forms relative to the non-phospho-C-tail form (*Figure 4* and *Figure 5B and C*). Similarly, the CST behavior for Gln43 and Asp44 was closely linked the phospho-state of the C-tail (*Figure 5G* and *Figure 5—figure supplement 3*) with an apparent loosening of the interaction for pSer473 but an apparent tightening for pSer477/pThr479. We therefore posit that this region plays a role in governing Akt regulation and investigated this possibility below.

## Role of a short hinge, aa 44–46, in Akt regulation

Prior X-ray crystal structures of the Akt PH domain have shown that aa 44–46 (DVD) located between β-strands 3 and 4 undergo a helix to loop transition between the inositol tetraphosphate-bound PH domain and the apo form (*Thomas et al., 2002*; *Milburn et al., 2003*; *Meuillet, 2011*; *Carpten et al., 2007*; *Figure 5—figure supplement 2F and G*). Given the fact that peaks assigned to Gln43 and Asp44 show either the highest CSP or extensive peak broadening as a function of C-tail phosphorylation status described above (*Figure 4*), we hypothesized that this conformational change contributes to Akt regulation in the intact protein. To test the hypothesis that this short hinge affects the capacity of Akt to bind PIP3, we recombinantly expressed a triple mutant – D44G/V45P/D46G – of the isolated PH domain aimed at disrupting the potential of this segment to form a stable helix with Gly and Pro residues that favor a random coil secondary structure (*Pace and Scholtz, 1998*). We then measured the affinity of wt and triple mutant PH domains to bind PIP3, using a fluorescence anisotropy assay; however, we did not detect any significant difference in the binding affinity (*Figure 6—figure supplement 1A*).

To characterize the effect of the short hinge further, we produced wt and triple mutant semisynthetic Akt variants using a standard expressed protein ligation strategy in which recombinant aa 1–459 thioester was generated in *Sf*9 insect cells along with co-overexpression with PDK1 to afford Thr308 phosphorylation. Using western blotting, we noted that the degree of phosphorylation at Thr308 was reduced compared to wt Akt prepared under the same conditions (*Figure 6—figure supplement 1B and C*). We thus treated partially Thr308 phosphorylated triple mutant (D44G/V45P/D46G) Akt1 aa 1–459 with PDK1 in an in vitro kinase reaction which appeared to complete the phosphorylation of Thr308 to a level matching that of wt Akt1 obtained simply from PDK1 co-expression

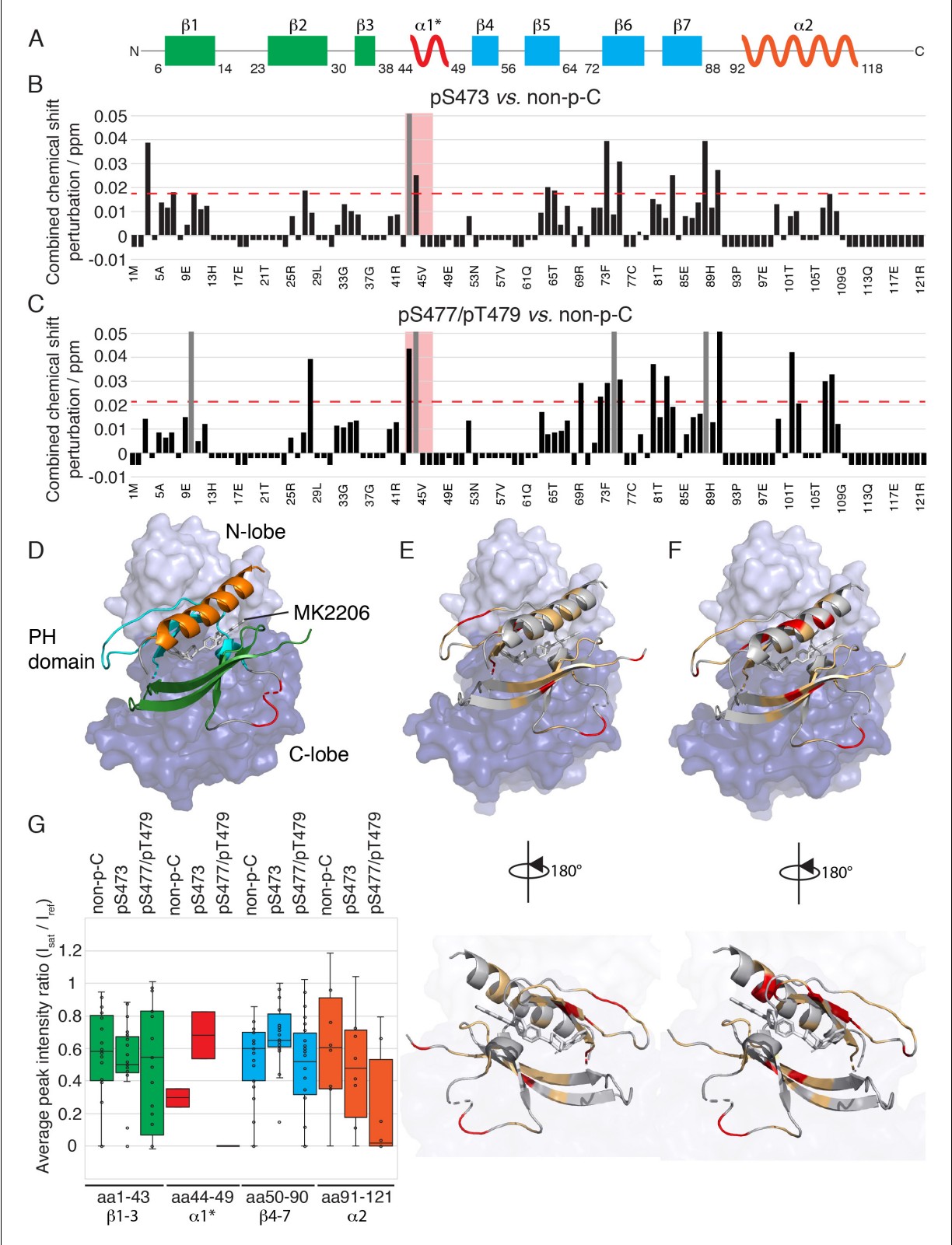

**Figure 5.** The PH domain of Akt interacts with the kinase domain differently depending on C-tail phosphorylations. (**A**) Cartoon representation of secondary structure elements (rectangle for β-strands, zigzag for α-helices) in Akt PH domain. Color coding represents regions with distinct binding modes to the kinase domain. Star indicates that the α-helix is present only when Akt is bound to IP4. (**B–C**) Combined chemical shift perturbations derived from spectra in *Figure 4* and plotted along the PH domain primary sequence for pSer473 (**B**) and pSer477/pThr479 (**C**) referenced to non-p-C. *Figure 5 continued on next page*

*Figure 5 continued*

Dashed red line corresponds to the standard deviation to the mean, excluding outliers (higher than 3xStDev). Grey bars indicate peaks that disappeared from the spectrum, also indicating strong interaction. Red area highlights the short hinge primarily studied here. Negative bars (−0.05) indicate non-assigned residues, negative bars (−0.025) indicate residues which assignment could not be easily transferred or recovered in the context of full-length Akt. (D) Structure of allosteric drug inhibited Akt (PDB: 3O96, [*Wu et al., 2010*]) with PH domain as ribbon in the front and kinase domain as surface in the back (N-lobe in light blue, C-lobe in dark blue). Color coding of secondary structure elements in the PH domain corresponds to (A). Allosteric inhibitor MK2206 is displayed in white. Main Akt domains are labeled. (E–F) Same structure representation as in (D) with the most significantly affected residues (chemical shift perturbations higher than the standard deviation) colored in red in the PH domain in case of pSer473 (E) and pSer477/pThr479 (F). Non-affected residues are shown in light orange and non-assigned residues in grey. Representations rotated by 180° are shown. (G) Statistical bar and whisker plots of saturation transfer efficiencies from CST data (*Figure 5—figure supplement 3*) for each C-tail phospho-state, categorized and color coded according to secondary structure elements as in (A) and (D). A ratio of 0 indicates maximum saturation transfer efficiency (very tight interaction) whereas a ratio of 1 indicate no saturation transfer (no interaction).

The online version of this article includes the following figure supplement(s) for figure 5:

**Figure supplement 1.** The PH domain of Akt is significantly affected in the context of full-length Akt.
**Figure supplement 2.** Akt kinase domain and inositol phosphates compete for interaction with PH domain.
**Figure supplement 3.** Cross-saturation transfer NMR analysis of the the PH-kinase domain interactions.

(*Figure 6—figure supplement 1E*). Achieving similar pThr308 levels for wt and D44G/V45P/D46G Akt is necessary to determine how the PH hinge mutation affects catalysis in the same context.

We then ligated Akt aa 1–459 recombinant thioesters with synthetic peptides containing aa 460–480 monophosphorylated at Ser473, diphosphorylated at Ser477 and Thr479, or non-phosphorylated (*Figure 6—figure supplement 1D*). Kinase assays performed with the triple mutant pSer473 full-length Akt semisynthetic protein showed that its catalytic efficiency, $k_{cat}/K_m$, is reduced ~10 fold relative to wt pSer473 under these conditions (*Figure 6A and B*). In contrast, the pSer477/pThr479 activation was essentially unaffected by the PH domain triple mutation (DVD - > GPG). The differences in sensitivity of the DVD triple mutation to the two C-tail phosphorylated forms are consistent with the distinct CST behaviors noted above (*Figure 5G* and *Figure 5—figure supplement 3*).

To examine the cellular significance of the D44G/V45P/D46G triple mutation, we transfected the full-length Akt triple mutant into an Akt1/Akt2 knockout HCT116 human colon cancer cell line. Cells were treated with insulin and IGF1 to stimulate the PI3K/Akt pathway that classically leads to phosphorylation of Thr308 as part of the activation mechanism. Interestingly, Thr308 phosphorylation was significantly impaired in the D44G/V45P/D46G triple mutant relative to transfected wt Akt (*Figure 6C*). These results are consistent with the insect cell expression experiments described above and suggest that the intramolecular PH domain-kinase domain interactions are altered in a way that interferes with PDK1-catalyzed phosphorylation of the activation loop.

To examine this in more depth, we performed an in vitro analysis of the kinetics of PDK1-mediated Thr308 phosphorylation of wt and GPG triple mutant pSer473 Akt. This revealed a ~ 2.5-fold reduction in Thr308 phosphorylation of the mutant protein, suggesting the reduced accessibility of Thr308 to PDK1 (*Figure 6D*).

## Structural changes in Akt induced by allosteric inhibitors

Crystal structures of near full-length Akt1 in complex with various allosteric inhibitors have been widely accepted to represent the inactive conformation of Akt, because the PH and kinase domains are glued together (*Wu et al., 2010*; *Calleja et al., 2009*; *Parikh et al., 2012*; *Lapierre et al., 2016*; *Ashwell et al., 2012*). In these inhibitor-bound structures, the PIP3-binding pocket of the PH domain is obstructed which would be expected to reduce PIP3-binding affinity. Our recent work has revealed that PIP3-binding affinity is similar for the pSer473 and non-C-tail phosphorylated Akt forms. In this light, it has been difficult to reconcile the structural studies. To investigate this further, we measured PIP3 affinity to Akt in the context of either MK2206 or compound VIII allosteric Akt inhibitors (*Figure 7*, *Figure 7—figure supplement 1A*). These experiments have shown that the allosteric inhibitors dramatically weaken the binding of PIP3 to Akt, by more than 20-fold (*Figure 7*).

We then used NMR to assess the impact of MK2206 on the conformation of the PH domain with non-C-tail phosphorylated Akt, where the PH domain was segmentally labeled. Based on the $^{15}$N-$^{1}$H HSQC analysis, there were marked chemical shift changes throughout the PH domain (*Figure 8A,D*). The amplitudes of these chemical shift perturbations were generally much larger than those

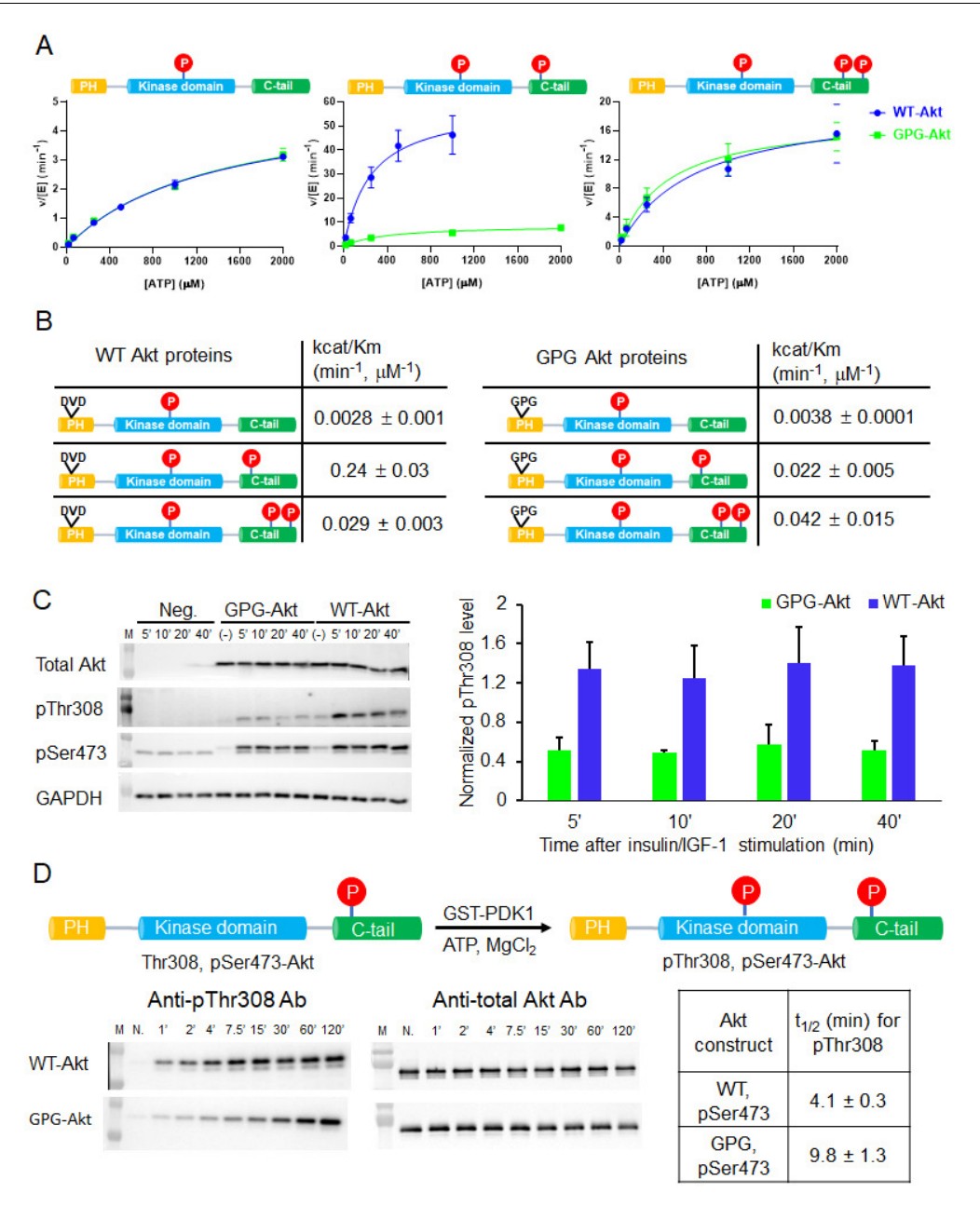

**Figure 6.** A short hinge 44–46 aa in the PH domain governs Akt activation using a two-pronged approach. (**A**) Steady-state kinetic plots v/[E] versus [ATP] with 20 µM GSK3 peptide for semisynthetic pThr308 Akt proteins WT (blue) versus D44G/V45P/D46G (GPG) mutant with non-P C-tail (left), pSer473 (middle) and pSer477/pThr479 (right), n = 2. (**B**) Catalytic efficiencies (apparent $k_{cat}/K_m$ values) for each semisynthetic Akt phospho form of WT (left) and GPG mutant (right) obtained from kinase assays in (**A**), two independent repeats were performed for each assay, S.D. shown. (**C**) Cellular analysis of the effect of GPG mutant on Akt phosphorylation. Left, Akt and GAPDH antibodies western blot of cell lysate at different time points after stimulation, as indicated, Neg.: non-transfected cells stimulated with insulin/IGF-1; (-): transfected with DNA plasmid but not stimulated with insulin/IGF-1 (left). Right, quantification of Akt Thr308 phosphorylation level (blue for WT and green for GPG mutant) using ImageJ2 (n = 5, SEM shown, 0.01 < p < 0.05). (**D**) Time course kinase assay for PDK1-catalyzed Akt Thr308 phosphorylation (schematic illustration). Left, western blot of semisynthetic pSer473 Akt (WT or GPG) after incubation with PDK1 for different times, as indicated. Right, corresponding calculated half-time of completion obtained from two independent repeats and S.D. shown.

The online version of this article includes the following figure supplement(s) for figure 6:

*Figure 6 continued on next page*

*Figure 6 continued*

**Figure supplement 1.** A short hinge 44-46 aa affects Thr308 phosphorylation but not PIP2 affinity.

observed when comparing different C-terminal phospho-forms of Akt. Notably, some of the most significant chemical shift perturbations were detected in the PIP3-binding interface (e.g. aa Trp11, Leu12) (*Figure 8D*). We also performed a CST experiment in the same manner as described above, in the presence of MK2206 (*Figure 8B,E and F*) and observed two principal effects mediated by the compound. The short hinge of aa 44–46 seems to separate further from the kinase domain and/or C-terminal tail while the C-terminal alpha-helix, in contrast, appears to interact more tightly with the kinase domain or C-tail. These inhibitor-induced changes likely result from a combination of altered structural and dynamic features relative to the baseline autoinhibited form of Akt. These could include a reorientation of the PH domain with respect to the kinase domain or C-tail (placing the C-terminal helix closer) as well as conformational and dynamic rearrangements within the PH domain itself. In a complementary approach, we performed a fluorescence trans binding assay of split Akt (separate PH and kinase domains) that MK2206 can enhance the binding affinity between the two domains in an intermolecular fashion (*Figure 7—figure supplement 1B and C*). This is consistent with a related study that showed that allosteric inhibitors indeed can stimulate kinase-PH domain interaction (*Parikh et al., 2012*; *Weisner et al., 2015*).

## Discussion

Our understanding of the molecular basis of Akt regulation by its PH domain have primarily derived from Akt crystal structures that either lack the PH domain or contain an allosteric inhibitor, although HD-exchange studies have also been performed on S473D Akt (*Lučić et al., 2018*). We note that prior studies have indicated that Asp473 is a poor mimic of pSer473 in kinase activity studies (*Chu et al., 2018*) suggesting that Asp473 may not relieve autoinhibition efficiently. Here, we look directly at the Akt PH domain in the context of full-length protein by solution NMR using a segmental labeling approach. These experiments show that specific regions in the PH domain show chemical shift perturbations in the presence of C-terminal phosphorylation that can stimulate Akt activity. Moreover, there are distinct changes in Akt that is phosphorylated on Ser473 versus Ser477/Thr479, consistent with differential interactions of the phospho-sites and the kinase domain and linker dependence on the phosphorylation state (*Chu et al., 2018*).

These studies highlight the value of segmental labeling using semisynthesis to make larger proteins dependent on insect cell expression amenable to NMR. The simplified spectra of the ~120 aa PH domain, prepared in *E. coli*, embedded in the 480 aa Akt protein, mostly produced in *Sf*9 cells, was straightforward to assign by correlation with the isolated PH domain. We suggest that such segmental labeling strategies be considered for solution phase structural studies of big proteins requiring eukaryotic expression that are now commonly considered out of reach by NMR.

We focused on one short segment in the PH domain based on the NMR studies which is the short hinge consisting of the three amino acid motif aa 44–46, DVD, henceforth referred to as the DVD motif. This hinge region was sharply affected by C-terminal phosphorylation. Prior crystal structures of the isolated PH domain have shown that this DVD motif transitions between alpha-helix and loop depending on whether the domains were complexed with the PIP3 mimic, IP4 (*Thomas et al., 2002*; *Milburn et al., 2003*; *Askham et al., 2010*; *Meuillet, 2011*). By mutating DVD to GPG to destabilize possible helix formation, we showed that C-terminal phosphorylation of Ser473 was poorly able to stimulate Akt kinase activity. This suggests that the normal dynamics (loop to helix transition) of this DVD motif is necessary for allowing the PH domain to dislodge from the kinase domain in response to pSer473. In contrast, activation of Akt by pSer477/pThr479 was insensitive to this mutation, despite changes in chemical shift in the hinge region (aa 43–44). These findings can be interpreted as consistent with a distinct mode of activation by these non-classical phosphorylation events (*Chu et al., 2018*; *Liu et al., 2014*). The GPG mutation in the isolated PH domain did not influence its ability to bind PIP3 indicating that the dynamics of this DVD segment is not intimately connected to the thermodynamics of PIP3 binding. Interestingly, our results show that the DVD motif can also

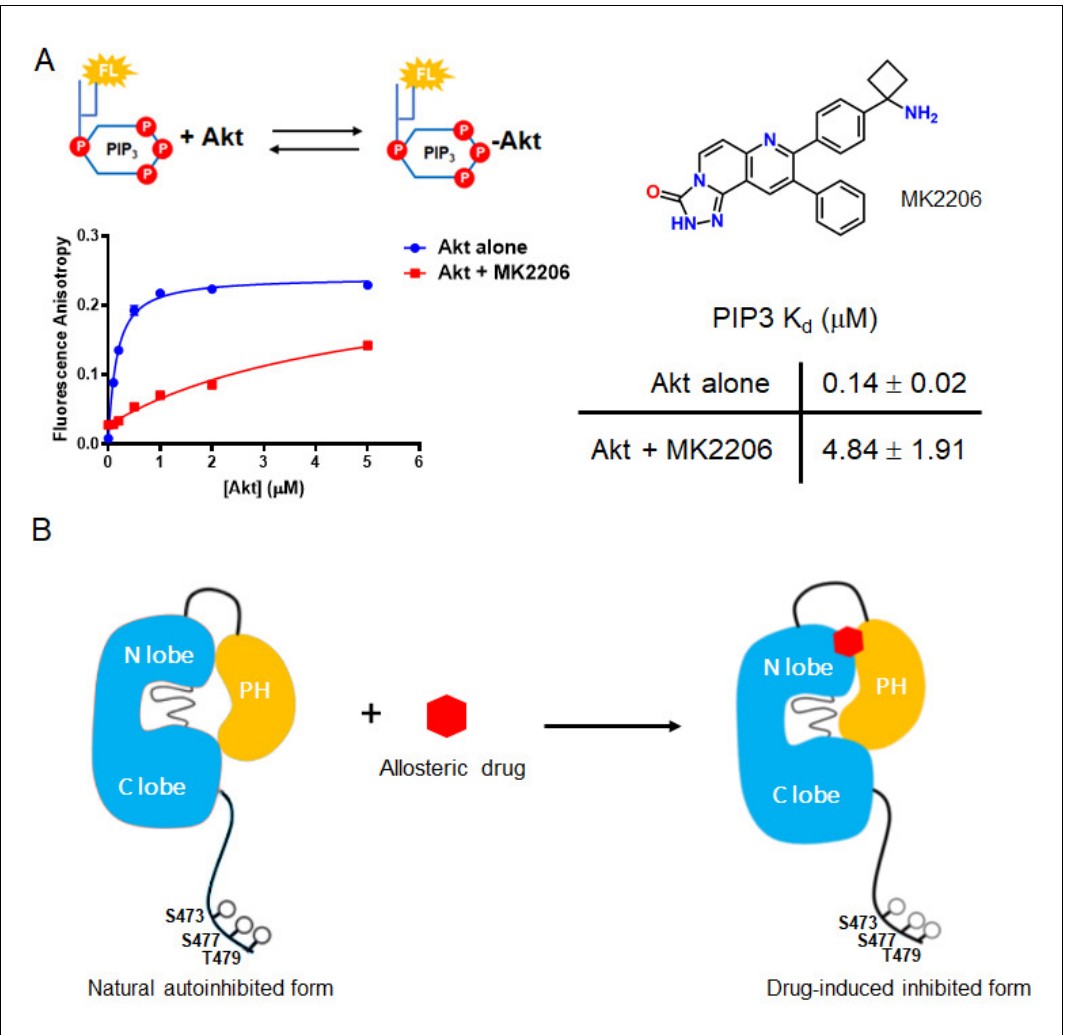

**Figure 7.** The allosteric drug-induced Akt inhibited form is distinct from its native autoinhibited form. (**A**) Binding assays of phospholipid PIP3 with full-length Akt in presence of 20 μM MK2206. Chemical structure of MK2206 is shown on the right. The fluorescence anisotropy measurements (n = 2) were carried out and fit to quadratic binding isotherms, and $K_d$ values shown ± S.D. (**B**) Cartoon model illustrated the distinct conformational structure of PH domain of allosteric drug-bound Akt when compared to that of natural autoinhibited form.
The online version of this article includes the following figure supplement(s) for figure 7:

**Figure supplement 1.** Allosteric inhibitors glue PH and kinase domains together and compete with PIP3.

---

impact PDK1's ability to phosphorylate Thr308 in Akt's activation loop. We surmise that the helical state of aa 44–46 in Akt may facilitate a catalytically competent interaction between PDK1 and Akt.

Our results also show how flexibility/length in the linker between the kinase domain and PH domain can modulate the ability of C-terminal phosphorylation to activate Akt. Insertion of a hexaGly linker led to reduced activation in the case of pSer473, but only minimally in the case of pSer477/pThr479. The results with the pSer473 form of Akt suggests that engagement of Arg144 by pSer473 may tug on the linker to weaken the PH domain-kinase domain interaction and that this effect is attenuated by increasing linker flexibility which could reduce linker tension (*Chu et al., 2018*). Prior cellular studies revealed that growth factor induced pSer473 phosphorylation is decreased in the hexaGly mutant, presumably because the C-tail phosphorylation is not as shielded from cellular phosphatases when the linker is less rigid. Notably, the baseline activity of non-C-tail-phosphorylated Akt is elevated in the hexaGly insertion mutant relative to the wt Akt enzyme. This

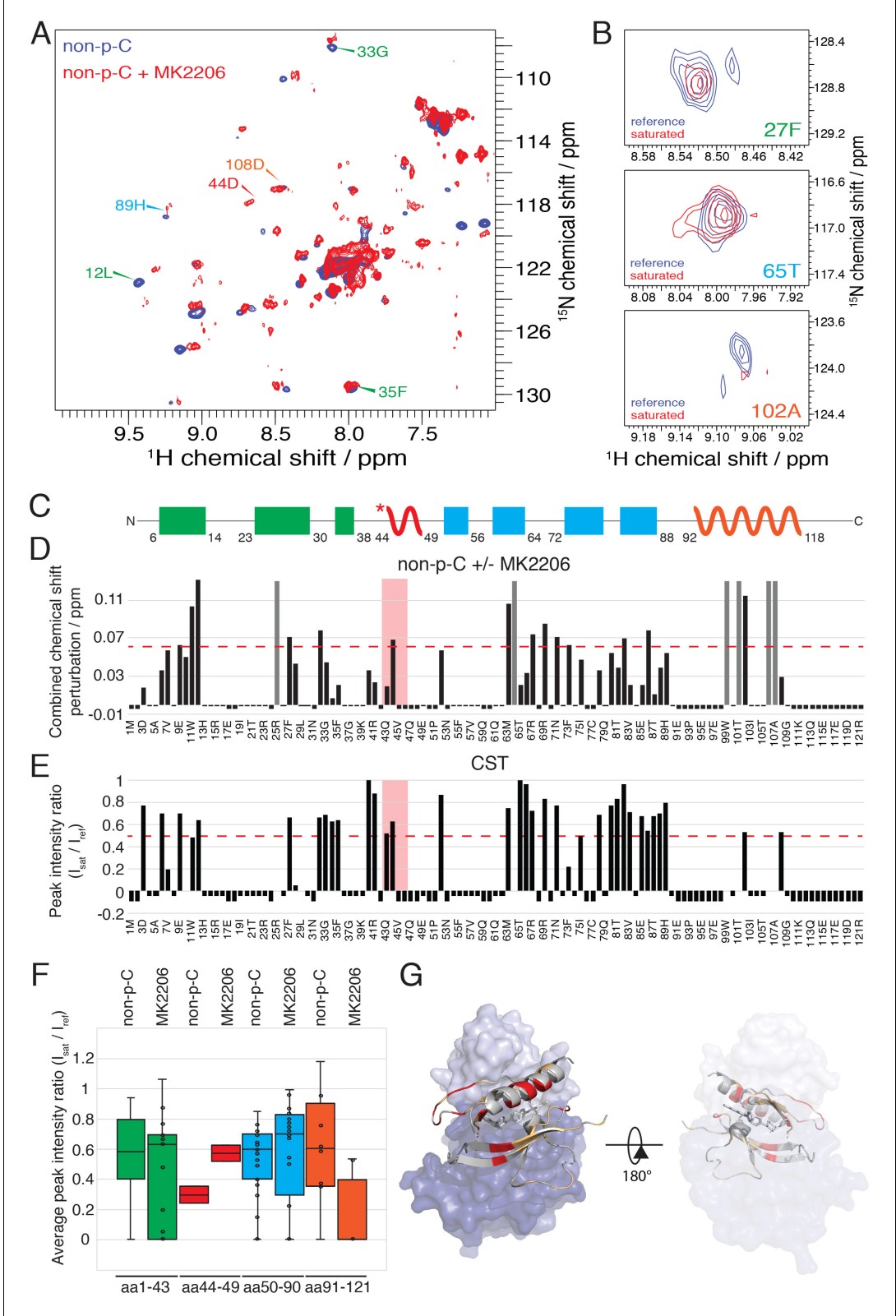

**Figure 8.** The allosteric drug-induced Akt inhibited form is distinct from its native autoinhibited form. (**A**) Overlay of $^{15}$N-$^{1}$H HSQC spectra of the PH domain in the context of full-length semisynthetic Akt with non-phosphorylated C-tail in the presence of the allosteric inhibitor MK2206 (red) or in its absence (blue). Select residue-specific assignments are shown. (**B**) Expanded $^{15}$N-$^{1}$H HSQC cross saturation transfer (CST) spectra around the peaks assigned to Phe27, Thr65 and Ala102. Reference (unsaturated) spectrum is shown in blue and its saturated counterpart in red. (**C**) Cartoon

*Figure 8 continued on next page*

*Figure 8 continued*

representation of secondary structure elements (rectangle for β-strands, zigzag for α-helices) in Akt PH domain. Color coding represents regions with distinct binding modes to the kinase domain. Star indicates that the α-helix is present only when Akt is bound to IP4. (D) Combined chemical shift perturbations corresponding to spectra in (A) plotted along the primary sequence for the PH domain in the context of semisynthetic full-length Akt with non-p C-tail with MK2206, referenced to the control without drug. Dashed red line corresponds to the standard deviation to the mean, excluding outliers (higher than 3xStDev). Grey bars indicate peaks that disappeared from the spectrum, also indicating strong interaction. Red area highlights the short hinge primarily studied here. Negative bars (−0.05) indicate non-assigned residues, negative bars (−0.025) indicate residues which assignment could not be easily transferred or recovered in the context of full-length Akt. (E) Saturation transfer efficiency derived from spectra in (B) plotted as the ratio of peak intensities of saturated over unsaturated spectra against the PH domain primary sequence. A ratio of 0 indicates maximum saturation transfer efficiency (very tight interaction), whereas a ratio of 1 indicate no saturation transfer (no interaction). An indicative dashed red line has been drawn at 0.5. Red area highlights the short hinge primarily studied here. Negative ratios (−0.05) indicate non-assigned residues, negative ratios (−0.025) indicate residues that were not present in 70% $^2$H$_2$O spectra. (F) Statistical bar and whisker plots of saturation transfer efficiencies from CST data (B,E), categorized and color coded according to secondary structure elements as in (C). (G) Structure of allosteric drug inhibited Akt (PDB: 3O96, [*Wu et al., 2010*]) with PH domain as ribbon in the front and kinase domain as surface in the back (N-lobe in light blue, C-lobe in dark blue). Significantly affected residues (chemical shift perturbations higher than the standard deviation) are colored in red. Non-affected residues are shown in light orange and non-assigned residues in grey. A representation rotated by 180° is shown.

can be understood to result from a floppier linker, enlarging the volume sampled by the PH domain, lowering its effective concentration relative to the kinase domain.

Our NMR analysis helps explain the paradox that crystal structures of allosteric inhibitors in complex with Akt show that the PH domain PIP3 pocket is obstructed but the non-phospho-C-tail form of Akt binds PIP3 potently. Indeed, we confirm here that in the presence of allosteric inhibitor, PIP3 affinity of Akt is markedly reduced. These results indicate that the conformation of the baseline non-phosphorylated-C-tail form of Akt is not identical to the allosteric-bound conformation. NMR data show dramatic chemical shift changes in the PH domain between the non-phosphorylated-C-tail and the inhibitor-bound form of Akt that likely reflect these differences. It is possible that the non-phosphorylated-C-tail Akt state is comprised of an ensemble of conformations, and one of the minor populations binds to allosteric inhibitors via conformational selection. Although NMR dynamics experiments, due to their very long measurement times are not feasible with our current experimental system, future alternative approaches may allow this to be investigated.

# Materials and methods

**Key resources table**

| Reagent type (species) or resource | Designation | Source or reference | Identifiers | Additional information |
|---|---|---|---|---|
| Gene (human) | Akt1 | Addgene and DOI:10.1093/nar/gkh238 | Addgene # 9021 | |
| Strain, strain background (*E. coli*) | Rosetta 2(DE3) pLysS | Novagen | Cat. No.: 71400 | Competent cells |
| Strain, strain background (*E. coli*) | DH10Bac | Invitrogen | Cat. No.: 18297010 | Competent cells |
| Cell line (insect cell) | Sf21 | Invitrogen | 11497–013 | |
| Cell line (insect cell) | f9 | Invitrogen | 11496–015 | |

*Continued on next page*

*Continued*

| Reagent type (species) or resource | Designation | Source or reference | Identifiers | Additional information |
|---|---|---|---|---|
| Cell line (human) | Akt1/2$^{-/-}$ HCT116 | DOI: 10.1073/pnas.0914018107 | | Colon cancer cell line; The Akt1$^{-/-}$ and Akt2$^{-/-}$ HCT116 colon cancer cell line was a gift from Dr. Bert Vogelstein (Johns Hopkins University) (*Ericson et al., 2010*). These cells were authenticated by western blot showing the absence of Akt and by the lack of signaling response to growth factors. They were also shown to be mycoplasma-free by PCR testing. |
| Antibody | pan-Akt (11E7) (Rabbit monoclonal) | Cell Signaling Technology | Cat. No.: 4685S, RRID:AB_10698888 | WB (1:1000 for cell-based assays; 1:20000 for activation assays) |
| Antibody | Akt phospho-Thr308 (D25E6) (Rabbit monoclonal) | Cell Signaling Technology | Cat. No.: 13038S, RRID:AB_2629447 | WB (1:1000, for cell-based assays; 1:10000 for activation assays) |
| Antibody | Akt phospho-Ser473 [EP2109Y] (Rabbit monoclonal) | AbCam | Cat. No.: ab81283, RRID:AB_2224551 | WB (1:1000) |
| Antibody | GAPDH (14C10) (Rabbit monoclonal) | Cell Signaling Technology | Cat. No.: 2118S, RRID:AB_561053 | WB (1:5000) |
| Antibody | HRP conjugated, anti-Rabbit IgG (Goat monoclonal) | Cell Signaling Technology | Cat. No.: 7074S, RRID:AB_2099233 | WB (1:5000) |
| Chemical compound, drug | Deuterium oxide (D, 99.8%) | Cambridge Isotope Laboratories | DLM-4–99.8-1000 | |
| Chemical compound, drug | Ammonium chloride ($^{15}$N, 99%) | Cambridge Isotope Laboratories | NLM-467–1 | |
| Chemical compound, drug | D-Glucose (U-13C6, 99%; 1,2,3,4,5,6,6-D7, 97–98%) | Cambridge Isotope Laboratories | CDLM-3813–2 | |
| Chemical compound, drug | Celtone base powder (13C, 98%+; D, 97%+; 15N, 98%+) | Cambridge Isotope Laboratories | CGM-1030P-CDN-1 | |
| Chemical compound, drug | D-Glucose (U-13C6, 99%) | Cambridge Isotope Laboratories | CLM-1396–2 | |
| Chemical compound, drug | 20 Fmoc-amino acids, Fmoc-Ser(HPO3Bzl)-OH, and Fmoc-Thr(HPO3Bzl)-OH | P3Bio systems | | |
| Chemical compound, drug | Sulfo-Cy5-NHS ester | Lumiprobe | Cat. No.: 43320 | |
| Chemical compound, drug | $^{32}$P-ATP | Perkin Elmer | Cat. No.: NEG002Z2-50UC | |
| Chemical compound, drug | Pierce avidin | Thermo Scientific | Cat. No.: 21128 | |

*Continued on next page*

*Continued*

| Reagent type (species) or resource | Designation | Source or reference | Identifiers | Additional information |
|---|---|---|---|---|
| Chemical compound, drug | Recombinant human insulin | Thermo Scientific | Cat. No.: 12585014 | |
| Chemical compound, drug | Human insulin-like growth factor 1 (hIGF-1) | Cell Signaling Technology | Cat. No.: 8917SC | |
| Software, algorithm | ImageJ2 | DOI: 10.1186/s12859-017-1934-z | | |
| Software, algorithm | GraphPad Prism version 8.2.1 | GraphPad | | |
| Software, algorithm | NmrPipe | DOI: 10.1007/BF00197809 | | |
| Software, algorithm | CCPNmr Analysis version 2.4 | DOI: 10.1002/prot.20449 | | |
| Software, algorithm | hmsIST | DOI: 10.1007/s10858-012-9611-z | | |
| Software, algorithm | Topspin version 3.6 | Bruker | | |

## Peptide synthesis

Peptides corresponding to residues 460–480 of Akt1 (CVDSERRPHFPQFSYSASGTA) and kinase substrate peptide N-ε-biotin-lysine GSK3 (RSGRARTSSFAEPGGK) were synthesized according to an Fmoc-based solid phase strategy previously described (*Chu et al., 2018*). The peptides were purified using reverse-phase C18 HPLC (VYDAC) using a gradient of 30% to 50% (v/v) of acetonitrile mixed with water containing 0.05% trifluoroacetic acid at a flow rate of 10 mL/min for 40 min. Pure fractions (>95%) were identified by MALDI mass spectrometry, combined and concentrated on a rotovap and then lyophilized to dryness. Peptide concentrations were determined by amino acid analysis.

## Expression of Akt proteins and GST-PDK1

The baculovirus-insect cell system was employed to express Akt-*Mxe*intein-CBD (CBD: chitin binding domain) constructs (Akt1 aa 2–459 and 122–459) and GST-PDK1 according to the procedures previously reported (*Chu et al., 2018*). In particular, the in vivo Akt Thr308 phosphorylation was obtained by co-expression of the Akt(2-459)-*Mxe*intein-CBD fusion construct with GST-PDK1 in Sf9 insect cells with M.O.I.s (multiplicity of infections) of 5.0 and 1.0 for the baculovirus containing Akt and GST-PDK1, respectively. After growing infected Sf9 cells for ~36 hr at 27°C, 25 nM of the phosphatase inhibitor okadaic acid (Cell signaling technology-CST) was added, and the cells permitted to grow for an additional 16 hr, and then harvested.

To obtain triply labeled $^{15}$N, $^{13}$C, $^{2}$H Akt PH domain, the pTXB1 plasmid containing Akt (aa 1–121)-*Mxe*Intein-CBD was expressed in *E. coli* Rosetta (DE3)/pLysS (Invitrogen) following the established protocol (*Gronenborn et al., 1991*; *Coote et al., 2018*). Briefly, the *E. coli* cells were grown in 1 L of M9 minimal medium (6 g/L $Na_2HPO_4$ (Sigma if not stated otherwise), 3 g/L $KH_2PO_4$, 0.5 g/L NaCl, 0.25 g/L $MgSO_4$, 11 mg/L $CaCl_2$, 2 g/L deuterated-$^{13}$C-glucose (Cambridge Isotopes), 1 g/L $^{15}NH_4Cl$ (Cambridge Isotopes), 100 mg/L ampicillin and 20 mg/L chloramphenicol) in $D_2O$, and was further supplemented with trace elements (50 mg/L EDTA, 8 mg/L $FeCl_3$, 0.1 mg/L $CuCl_2$, 0.1 mg/L $CoCl_2$, 0.1 mg/L $H_3BO_3$, and 0.02 mg/L $MnCl_2$) and the vitamins biotin (0.5 mg/L) and thiamin (0.5 mg/L) in shaker flasks at 37°C until $OD_{600}$ = 0.5, then 1 mL of 0.5 M IPTG was added to induce the expression and the cultures were further incubated for 24 hr at 16 °C. Cells were pelleted and stored in −80 °C freezer for the next steps.

## Semisynthesis of segmentally isotopically labeled Akt

To produce full-length Akt containing segmentally triply labeled $^{15}$N, $^{13}$C, $^{2}$H PH domain and the C-tail site-specific phosphorylations at either Ser473, Ser477/Thr479 or no phosphorylations on these residues, a sequential expressed protein ligation (EPL) strategy involving three peptide/protein

pieces was developed. After resuspending the *E. coli* cells expressing isotopically labeled PH domain-*Mxe*Intein-CBD in lysis buffer (50 mM HEPES pH 7.5, 150 mM NaCl, 1 mM EDTA, 10% Glycerol, 0.1% Triton X-100, one protease inhibitor tablet (Roche)), the cells were lysed by french press and the mixture was clarified by centrifugation at 17,500 g for 40 min at 4°C. The unlabeled insect cells expressing Akt (aa122-459-*Mxe*Intein-CBD) were suspended in lysis buffer and lysed in a 40 ml Dounce homogenizer on ice, and the mixture was clarified as described above for the PH domain. The insect cell expressed protein was also passed through fibrous cellulose to remove chitinase as described previously (*Bolduc et al., 2013*). Next, both N-Tags-TEV-S122C-Akt kinase domain (aa 122–459)-*Mxe*Intein-CBD (N-tags: N-terminal Flag-HA-6xHis) and triply labeled Akt PH domain (aa 1–121)-*Mxe*Intein-CBD proteins were purified by affinity chromatography from the cell lysates using chitin beads. After loading onto chitin beads, elution of the protein C-terminal thioester forms of both the Akt kinase and PH domains via intein cleavage using MESNA (sodium mercaptoethylsulfonate) according to established protocols (*Chu et al., 2018*). The obtained N-Tags-TEV-S122C-Akt kinase domain thioester was phosphorylated at Thr308 in vitro using recombinant GST-PDK1 (*Chu et al., 2018*), and then ligated with the synthetic N-Cys containing C-terminal Akt peptides (aa 460–480) containing variable phosphorylations in the first ligation buffer (50 mM HEPES pH 7.5, 150 mM NaCl, 1 mM TCEP, 100 mM MESNA, 10 mM EDTA, 10% glycerol, 1 mM PMSF) for 5 hr at room temperature and then maintained overnight at 4°C. The ligation product N-Tags-TEV-S122C-Akt aa 122–480 fragment was purified by size exclusion chromatography (SEC) on a Superdex 75 10/300 GL column (GE Healthcare) with the second ligation buffer (100 mM HEPES pH 7.8, 500 mM NaCl, 100 mM MESNA, 0.5 mM IP6 (phytic acid sodium salt, Sigma), 1 mM TCEP, 1 mM PMSF). The purified fractions (>90%) were combined, concentrated and mixed with triply labeled Akt PH domain thioester at the concentration of 5 mg/mL, 300 µg of TEV protease was added to initialize the second EPL at 4 °C for 65 hr. The ligation yields were assessed by coomassie-stained SDS-PAGE and typically shown to be greater than 90%. The resulting full-length Akt (aa 1–480) was separated from the excess of Akt PH domain thioester by using SEC purification on a Superdex 200 10/300 GL column (GE Healthcare) and buffer (50 mM HEPES pH 7.5, 150 mM NaCl, 2 mM beta-mercaptoethanol, 0.5 mM PMSF, 10% glycerol), concentrated, fast frozen and stored at −80°C.

Semisynthetic Akt proteins (wt, hexaGly, DVD mutant) used for kinase assays were generated by the previously described two-piece EPL strategy (co-overexpression with GST-PDK1 and okadaic acid treatment during insect cell expression) with the intein-mediated Akt (aa 1–459)-thioester fragment was reacted with the synthetic N-Cys containing C-terminal Akt peptides (aa 460–480) in the first ligation buffer as described above to afford semisynthetic full-length Akt proteins. The semisynthetic Akt proteins were purified by SEC with the Akt purification and storage buffer. The small contamination (~10%) of slightly truncated Akt proteins in these likely results from incomplete ligation or slight proteolytic cleavage of the ca. 40 aa N-terminal tag. As the C-terminal tail is critical in supporting Akt catalytic activity and the absence or presence of the N-terminal tag does not appear to affect catalytic activity (*Chu et al., 2018*), we do not think this contamination will have a big impact on the measured catalytic efficiencies.

## NMR experiments

All NMR experiments were conducted on a Bruker spectrometer operating at 750 MHz, equipped with a TCI cryoprobe and z-shielded gradients. All data were processed using Topspin (Bruker) or NmrPipe (*Delaglio et al., 1995*) and analyzed with CCPNmr (*Vranken et al., 2005*). Samples of approximately 600 µM $^2$H, $^{13}$C, $^{15}$N-labeled Akt PH domain (aa 1–121) in 50 mM HEPES pH 6.5, 500 mM NaCl, 2 mM IP6, 0.5 mM TCEP and 5% v/v $^2$H$_2$O were prepared for backbone resonance assignment (*Auguin et al., 2004*). Standard triple resonance backbone experiments: HSQC, HNCA, HNCOCA, HNCO, HNCACO, HNCACB were used. For all the backbone triple resonance experiments, 10% Nyquist grid, in the indirect dimension, was sampled using Poisson-Gap sampling. The resulting non-uniformly sample spectrum was reconstructed using the hmsIST (*Hyberts et al., 2012*). The NMR experiments were performed at 20°C. An additional HNCA spectrum was measured at 13°C, as well as a temperature titration (from 13°C to 20°C), allowed us to leverage the previously published assignments from *Auguin et al., 2004*. 66% of non-proline backbone resonances could be assigned.

HSQC spectra of 100–120 µM segmentally isotopically labeled full-length Akt were measured in 50 mM HEPES pH 6.5, 50 mM NaCl, 0.5 mM TCEP (NMR buffer) and 5% v/v $^2$H$_2$O using TROSY-

HSQC (*Pervushin et al., 1997*). Recycle delays of 1 s were used and 768 transients (and 256 increments in the indirect dimension) were accumulated, leading to measurement times of 54 hr. Chemical shifts were calibrated using internal water calibration in NmrPipe. Combined chemical shift perturbations and associated standard deviations were calculated according to the protocol from *Schumann et al., 2007*.

Cross saturation transfer experiments (*Shimada, 2005*) were measured with on (0 ppm) and off-resonance saturation (−30 ppm) in an interleaved fashion using 100–120 μM samples in NMR buffer. The samples were prepared in 70% v/v $^2H_2O$ to minimize spin-diffusion through solvent. Saturation time was set to 3 s and recycle delays to 1 s. Composite WURST pulses (25 milliseconds in duration) were used for saturation. Number of transients was approximately 400 and number of increments to 232 leading to measurement time of about 100 hr.

## Kinase activity assays

The steady-state kinetic parameters of semisynthetic Akt proteins were determined by phosphorylating N-ε-biotin-lysine GSK3 peptide (RSGRARTSSFAEPGGK) in radiometric reactions as described elsewhere (*Qiu et al., 2009*; *Chu et al., 2018*), with minor modification. The biotinylated GSK3 peptide substrate concentration was kept at 20 μM (5–10-fold above the peptide substrate $K_m$ values for Akt protein phosphorylated on Thr308 [*Chu et al., 2018*]) in all the assays. The buffer used contains 50 mM HEPES pH 7.5, 10 mM $MgCl_2$, 1 mM EGTA, 2 mM DTT, 1 mM sodium orthovanadate, 0.5 mg/mL BSA (kinase reaction buffer),~0.42 μCi γ-$^{32}$P-ATP, and varying amounts of ATP (0–2 mM). All the Akt proteins used in this assay contain N-terminal affinity tags Flag, HA and 6xHis unless otherwise noted. The kinase reactions were performed at 30°C for 10 min and quenched by adding 50 mM EDTA, and then 100 μg of Avidin (Pierce) was added to each sample and incubated for 20 min at room temperature. Samples were transferred to centrifugal 10 kDa MWCO filtration units (Nanosep 10K, PALL) and washed five times with 120 μl of washing buffer (0.5 M sodium phosphate, 0.5 M NaCl, pH 8.5). The filtration units were placed in 5 mL scintillation fluid and counted by Beckman liquid scintillation counter (Beckman LS6500). The $k_{cat}/K_m$ (ATP) values were calculated using the standard Michaelis-Menten equation using a non-linear fit with Prism (TM) and can be considered 'apparent' values performed at a fixed, although most likely near-saturating peptide substrate concentration (*Chu et al., 2018*).

## Fluorescence anisotropy measurements

The binding affinity of Akt PH domain with phospholipid PIP3 was determined using fluorescence anisotropy. Varying amounts of Akt PH domain were mixed with 50 nM fluorescein-labeled soluble PIP3 (Cayman Chemical) in binding buffer (50 mM HEPES pH 7.5, 2 mM DTT, 0.05 mg/ml ovalbumin) and incubated at room temperature for 30 min. Fluorescent anisotropy spectra were recorded by Multi-Mode Microplate Reader (Biotek Instruments) at 23°C with three different replicates. The $K_d$ values were obtained by fitting the data to quadratic binding equation as described before (*Seamon et al., 2015*; *Weiser et al., 2017*; *Chu et al., 2018*).

## Microscale thermophoresis (MST) analysis

For use in MST analysis, N-terminally Cy5 labeled, PH-deleted S122C Akt (aa 122–480) was prepared by pretreating Sulfo-Cy5-NHS ester (Lumiprobe) with MESNA to efficiently convert NHS ester into thioester that can selectively react with N-terminal Cysteine 122 as described in *Dempsey et al., 2018*. The binding affinity of Cy5 labeled, S122C Akt (aa 122–480) and Akt PH domain (aa 1–121) in a binding buffer containing 50 mM HEPES pH 7.5, 150 mM NaCl, 5% (v/v) glycerol, 0.05% (v/v) Triton X-100, 0.1 mg/mL ovalbumin, 5 mM DTT was carried out by MST using MONOLITH NT.115 (NanoTemper). The Cy5-labeled Akt (aa 122–480) (20 nM) was mixed 1:1 with different amounts of Akt PH domain in a two-fold dilution series from 19.8 nM to 650,000 nM for the measurement at 23° C. For the binding assays with the presence of Akt allosteric inhibitor MK2206, 20 nM Cy5-labeled Akt (aa 122–480) was pre-incubated with 40 μM MK2206 on ice for 30 min and then mixed 1:1 with Akt PH domain as described above. Each binding assay was repeated twice.

## PDK1 phosphorylation of Akt assays

The PDK1 phosphorylation of Akt assays were carried out following the protocol previously described (*Chu et al., 2018*). Briefly, 1 µM Akt was mixed with 10 nM GST-PDk1 in the activation buffer (50 mM HEPES, pH 7.5, 2 mM DTT, 10 mM $MgCl_2$) and incubated at 30°C, the reaction was triggered by adding 1 mM ATP. 10 µl of the reaction was collected at the indicated time points and quenched by the addition of 4xSDS-loading buffer. The SDS samples were loaded on SDS-PAGE gel for western blot analysis with anti-pT308 or pan-Akt primary antibodies.

## Western blots

After transferring protein from SDS-PAGE gels to nitrocellulose membranes using an iBlot (Thermo Fisher) system, the membranes were blocked with 5% (w/v) BSA in TBS-T buffer at room temperature for 30 min. Membranes were incubated with anti-Akt (pan or phospho) primary antibodies (CST) at a 1:10,000 dilution with 5% (w/v) BSA in TBS-T buffer overnight at 4°C, washed three times of 10 min with TBS-T buffer, and incubated with secondary HRP-linked antibody (CST) for 1 hr at room temperature and following with three washing times. Membranes were developed with Amersham ECL Western blotting detection reagents (GE Healthcare) and imaged by a GeneSys (G:BOX, Syn-Gene) imaging system.

## Mammalian cell signaling assays

The Akt1$^{-/-}$ and Akt2$^{-/-}$ HCT116 colon cancer cell line was a gift from Dr. Bert Vogelstein (Johns Hopkins University) (*Ericson et al., 2010*). These cells were authenticated by western blot showing the absence of Akt and by the lack of signaling response to growth factors. They were also shown to be mycoplasma-free by PCR testing. Cells were cultured in McCoy's 5A (Invitrogen) supplemented with 10% (v/v) FBS (Sigma) and 1% (v/v) penicillin/streptomycin (Gibco) at 37°C and 5% $CO_2$. When the cell reached ~70% confluence in six-well plates, the cells were transfected with 1 µg of pcDNA3.1-Flag-HA-Akt plasmids complexed with 2 µL Lipofectamine 3000 (Invitrogen) and 2 µL P3000 reagent (Invitrogen) in Opti-MEM medium (Gibco) for 24 hr at 37°C and 5% $CO_2$. When indicated, the cells were rinsed twice by PBS and serum-starved for 18 hr in McCoy's 5A with 0.5% FBS and 1% penicillin/streptomycin, and stimulated with 100 ng/mL of insulin (Thermo Fisher Scientific) and 60 ng/mL human IGF-1 (CST) for variable times (5, 10, 20, and 40 min) at 37°C and 5% $CO_2$. The cells were lysed by adding 180 µL RIPA buffer (CST) containing 1x complete protease inhibitor tablet and 1 mM PMSF, and gently shook for 30 min at 4 °C. 50 µg of total protein (BCA assay) was loaded on SDS-PAGE gels. Membrane transfer and Western blotting was carried out as described above with 1:1000 dilution for primary antibodies: Akt1, phospho Akt1.

## Acknowledgements

We thank TK Harris for helpful advice and encouragement. HA acknowledges funding from the Claudia Adams Barr Program for Innovative Cancer Research. AB was supported by an Austrian Science Fund's Schrödinger Fellowship (J3872-B21) and an American Heart Association's fellowship (19POST34380800). Maintenance of the NMR instruments used for this research was supported by NIH grant no. EB002026. PAC and AS are grateful for funding from NIH grant CA74305. HB thanks the Kwanjeong Educational Foundation for support.

## Additional information

### Competing interests

Philip A Cole: Senior editor, *eLife*. The other authors declare that no competing interests exist.

### Funding

| Funder | Grant reference number | Author |
|---|---|---|
| National Cancer Institute | CA74305 | Philip A Cole |

| | | |
|---|---|---|
| Claudia Adams Barr Program in Innovative Cancer Research | | Haribabu Arthanari |
| Austrian Science Fund | Schroedinger Fellowship | Andras Boeszoermenyi |
| American Heart Association | 19POST34380800 | Andras Boeszoermenyi |
| National Institutes of Health | EB002026 | Haribabu Arthanari |
| Kwanjeong Educational Foundation | Pre-doctoral fellowship | Hwan Bae |

The funders had no role in study design, data collection and interpretation, or the decision to submit the work for publication.

## Author contributions

Nam Chu, Thibault Viennet, Conceptualization, Data curation, Formal analysis, Investigation, Writing - original draft, Writing - review and editing; Hwan Bae, Antonieta Salguero, Data curation, Formal analysis, Investigation, Writing - review and editing; Andras Boeszoermenyi, Formal analysis, Investigation, Writing - review and editing; Haribabu Arthanari, Conceptualization, Resources, Data curation, Formal analysis, Supervision, Investigation, Writing - original draft, Project administration, Writing - review and editing; Philip A Cole, Conceptualization, Supervision, Writing - original draft, Project administration, Writing - review and editing

## Author ORCIDs

Nam Chu http://orcid.org/0000-0001-9717-5007
Thibault Viennet https://orcid.org/0000-0001-5349-0179
Hwan Bae http://orcid.org/0000-0001-5252-252X
Philip A Cole https://orcid.org/0000-0001-6873-7824

## Decision letter and Author response

Decision letter https://doi.org/10.7554/eLife.59151.sa1
Author response https://doi.org/10.7554/eLife.59151.sa2

# Additional files

## Supplementary files

• Transparent reporting form

## Data availability

Source data: Enzyme kinetics, fluorescence binding data, western blots, SDSPAGE gels have been deposited in Dryad: https://doi.org/10.5061/dryad.0p2ngf1xg.

The following dataset was generated:

| Author(s) | Year | Dataset title | Dataset URL | Database and Identifier |
|---|---|---|---|---|
| Cole PA, Chu N, Viennet T, Bae H, Salguero A, Boeszoermenyi A, Arthanari H | 2020 | The Structural Determinants of PH Domain-Mediated Regulation of Akt Revealed by Segmental Labeling | https://doi.org/10.5061/dryad.0p2ngf1xg | Dryad Digital Repository, 10.5061/dryad.0p2ngf1xg |

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
