## [Decision Letter]

Thank you for submitting your article "The Structural Determinants of PH Domain-Mediated Regulation of Akt Revealed by Segmental Labeling" for consideration by *eLife*. Your article has been evaluated favorably by three peer reviewers, and John Kuriyan has overseen the evaluation as the Reviewing and Senior Editor. The following individuals involved in the review of your submission have agreed to reveal their identity: Kevin N. Dalby (Reviewer #1); Neel Shah (Reviewer #3).

The reviewers have discussed the reviews with one another, and the Reviewing Editor has drafted this decision to help you prepare a revised submission.

The manuscript by Cole, Arthanari, and co-workers aims to enhance our understanding of activation and autoinhibition of the protein kinase Akt1 (AKT). The authors use a combination of protein semisynthesis, enzymological assays, and NMR spectroscopy to identify structural changes that distinguish variously autoinhibited and activated states of AKT.

Three particularly notable conclusions stand out from this work. First, the authors lay out an elegant route to making segmentally-isotopically labeled and site-specifically phosphorylated samples of a multi-domain protein that has mostly eluded NMR studies, thus far. Second, the NMR experiments with different phospho-forms of AKT, which could only have been obtained through their synthetic methods, substantiate the hypothesis that distinct phosphorylation events activate AKT via different mechanisms. Finally, their NMR data show that the auto-inhibited state of AKT observed in crystal structures is almost certainly a drug-induced conformation, and is unlikely to be the predominant structure in solution in the absence of an allosteric inhibitor. The study provides good evidence that the conformational dynamics of the linker (or "tension", but note that reviewers object to this word) between the PH domain of AKT and the kinase domain plays a role in mediating the activation of AKT by phosphorylation of Ser473. Moreover, it suggests this is not the case for the phosphorylation of Ser477/Thr479. The authors identify a small region within the PH domain that adopts a helical state and mediates the phosphorylation of Thr308 by PDK1 in intact cells.

The work presented here provides new methods to study kinase structure/function, and the results are likely to guide future biological investigations, and drug discovery efforts focused on AKT.

The review process did perceive weaknesses in the manuscript. Some aspects of the work may be considered incremental, as it focuses on linker length (which was partly examined in previous work from this group) and a single loop region in the PH domain. Given the tools at this group's disposal, it seems the data in this paper are underutilized, and in some cases, the experimental design seems ill-conceived (see comment #5 below). There is a concern that, as presented, the authors provide interesting observations, but the data are not interpreted in a manner that yields a definitive advance in our mechanistic understanding of AKT regulation.

Despite these weaknesses, the reviewers agreed that the impressive demonstration of segmental labeling to probe the effects of phosphorylation and to investigate PH domain conformation are important advances. The finding that the drug-induced conformation of Akt may not represent the physiological conformation is also important. The paper should become acceptable for publication in *eLife* once it is revised to take into account the comments below. Note that under the present circumstances we are only asking for textual revisions, but if additional data that support the claims are available, they could be included.

Comments to address:

1) Work by Lucic et al. https://doi.org/10.1073/pnas.1716109115 contains important contributions to understanding AKT regulation and yet is not described or referenced in the submitted manuscript. Please include it in the revised manuscript.

2) The authors suggest that the putative regulatory interaction between pSer473 and R144 "exerts tension on the PH-kinase domain linker, thereby pulling on the PH domain and severing its contacts…" The use of the word "tension" throughout is misleading as the authors have not measured force in any of their experiments. Instead, it seems more appropriate to think about the pSer473/R144 interaction as promoting a shift in the conformational ensemble of the full-length protein that leads to destabilization of the PH/kinase interface.

3) The authors claim the results of increasing linker length (addition of the hexaGly insert) "support a role for linker tension," but really linker length has been probed. A simple explanation may be that the increased linker length and flexibility in the region adjacent to R144 might create an increased entropic cost in forming the pSer473/R144 contact. There seems to be an assumption in these experiments that the pSer473/R144 interaction is unchanged upon the insertion of hexaGly.

4) The NMR data, as presented in Figure 4 are difficult to see. Perhaps the spectra could be expanded to show key spectral changes. More importantly, the NMR changes are never mapped on the PH domain structure. Figure 4F simply shows regions of secondary structure, but the data itself (Figure 4D and E) could be significantly more informative if mapped onto the PH domain structure. This is true for all of the HSQC data throughout the paper.

5) What are the phosphorylation states at T308 for all of the protein samples generated via semisynthesis? It appears that Thr308 is phosphorylated in the protein samples prepared for the NMR work. This is puzzling as the authors are probing the transition from autoinhibited to activated AKT via C-tail phosphorylation. So it seems that as long as the activation loop is phosphorylated they won't capture the fully autoinhibited form. Moreover, the scheme in Figure 1B indicates that C-tail phosphorylation occurs before activation loop phosphorylation providing another reason to study C-tail phosphorylation effects in the absence of activation loop phosphorylation. Please clarify and comment on this issue. Can the authors provide MS data confirming the identities (and purity) of all semisynthetic products generated in this study? If additional data supporting the analysis are available, it might be worth including in the manuscript.

6) The structural distinction between the natural auto-inhibited state and the drug-induced state is interesting but poorly defined here. A potentially important finding is that the AKT crystal structures containing small molecules that “staple” the PH domain to the kinase domain may not be revealing the native autoinhibitory interface. The NMR mapping in this paper could provide insight into the PH/kinase interface in the absence of drug, but the authors stop short of using their data in this way. Is there some reason that the data have not been pushed to provide a more definitive understanding of the PH-kinase interaction? It would be illustrative and helpful to see the residues with significant chemical shift perturbations in Figure 6 highlighted on the structure of the drug-bound autoinhibited state. A discussion of how the extended interface upon drug binding differs from the native autoinhibited state is warranted. What can be inferred from the cross saturation transfer experiments? Was such an experiment done in the presence of the allosteric inhibitor? This experiment isn't necessary, but a general discussion of structural models that can be inferred from the data in this manuscript would strengthen the manuscript.

7) Cross-saturation transfer NMR data are also underutilized. Examining the quantitative differences between non-p-C and pS475 should provide more precise insight into how serine phosphorylation might be affecting the conformational preference of the PH domain. It's difficult to see from Figure 4—figure supplement 3 how the cross-saturation transfer effect is stronger for non-phosphorylated versus tail-phosphorylated samples, as stated in the text. Could this be quantified in some way – perhaps showing the mean peak intensity ratios for each sample or a distribution of these ratios across all samples?

8) The heavy focus on the loop following b3 (residues Gln43 and Asp44) seems driven (according to the text) by more dramatic chemical shift perturbation and more extensive line broadening in the phosphorylated forms compared to non-p-C. This seems a stretch as there are certainly other regions that show equally or nearly the same extent of spectral change (it's possible that the HSQC data itself makes the differences more evident, but expanded spectral data showing individual HSQC peaks are not provided).

9) The remainder of the paper focuses on a mutation of the native DVD sequence (residues 44-46) to GPG in an effort to "favor a random coil secondary structure in this region." The authors do not characterize this mutant to determine whether the mutation successfully shifted secondary structure preference of the PH domain but instead put the mutation directly into their full-length semi-synthetic AKT. Despite the lack of characterization of the mutant, this is the one place in this paper where the authors see a dramatic result; Figure 5A shows that the mutation of the DVD motif causes a large decrease in the steady-state kinetics of AKT in its pSer473 form compared to wild type. The other phosphorylated form of AKT (pSer477/pThr497) is unaffected by the mutation in the PH domain. Clearly understanding the mechanistic basis for this difference would be an advance in this field but the manuscript as presented does not sufficiently explain the observations.

10) In the Discussion, the authors interpret the failure of pSer473 to stimulate AKT activity as being due to changes in the "normal dynamics" of the DVD motif. Dynamics can be directly measured by NMR to support this conclusion, but this is surprisingly not included in the manuscript. The observation that activation loop phosphorylation (at Thr308) is also diminished upon mutation of the DVD motif to GPG might suggest that the mutation in the PH domain enhances the autoinhibitory contact between PH and kinase domains and/or alters the accessibility of the activation loop to PDK-1. Unfortunately, as presented, the manuscript provides little to advance our understanding of the precise role of this region of the PH domain in AKT regulation.

11) In the final part of this work the authors explore the effects of drug binding on PIP3 binding as well as PH domain NMR chemical shifts. The data are not interpreted in a satisfactory manner; the authors state that "Notably, some of the most significant chemical shift perturbations were detected in the PIP3-binding interface…" but there are significant changes elsewhere in the PH domain as well. There is also no attempt to take into account the likely significant spectral changes that will arise from the drug itself.

12) Finally, the last sentence of the paper suggests conformational heterogeneity might be present in the non-phosphorylated C-tail form of AKT. This is very logical and raises the question of whether the authors see any evidence for the existence of multiple states in solution for any of the segmentally labeled forms they have studied in this submitted paper or the previous 2018 publication?

13) Some of the kinetic parameters for the WT semi-synthetic AKT proteins shown in Figure 2 seem to differ a little from the parameters reported earlier in Chu et al., 2018. For example, the kcat/Km for A2 (full-length AKT phosphorylated on T308) in Figure 2 of Chu et al. is 28-fold smaller than the value reported in the current manuscript. In the present manuscript, the concentration range of ATP used is 0-2000 µM, whereas the Km values reported for A2, A1, and A6 (Figure 2 of Chu et al.) are 200, 5100, and 3700 µM, respectively. Perhaps the authors could clarify the apparent discrepancy.

14) Perhaps mention that the reported kcat/Km's are apparent values as the second substrate (the peptide in this case) is not present at a saturating concentration.

15) It would be helpful in Figure 1 if the various states were labeled as "active" or "inactive." In Figure 1B, the ribbon diagram of the PH-kinase unit is drawn upside. In this and other figures (e.g., 4F), it would be helpful if structural components and parts were clearly labeled on the figures.

16) Can the authors comment on the purity of the samples used in Figure 2? Based on the figure supplement SDS-PAGE analysis, there appears to be a ~10-20% impurity. Can this be identified by MS analysis of the full-length protein to confirm that it isn't a species that will interfere with the analysis of enzyme kinetics?

---

## [Author Response]

The work presented here provides new methods to study kinase structure/function, and the results are likely to guide future biological investigations, and drug discovery efforts focused on AKT.The review process did perceive weaknesses in the manuscript. Some aspects of the work may be considered incremental, as it focuses on linker length (which was partly examined in previous work from this group) and a single loop region in the PH domain. Given the tools at this group's disposal, it seems the data in this paper are underutilized, and in some cases, the experimental design seems ill-conceived (see comment #5 below). There is a concern that, as presented, the authors provide interesting observations, but the data are not interpreted in a manner that yields a definitive advance in our mechanistic understanding of AKT regulation.Despite these weaknesses, the reviewers agreed that the impressive demonstration of segmental labeling to probe the effects of phosphorylation and to investigate PH domain conformation are important advances. The finding that the drug-induced conformation of Akt may not represent the physiological conformation is also important. The paper should become acceptable for publication in eLife once it is revised to take into account the comments below. Note that under the present circumstances we are only asking for textual revisions, but if additional data that support the claims are available, they could be included.

We agree that the strengths of our manuscript are the combination of the technical novelty as well as the mechanistic insights in Akt regulation and inhibition. We readily concede that our studies have not fully illuminated the complete set of conformational changes and altered intramolecular interactions in the various Akt states. A fundamental limitation of our work is the absence of NMR analysis of the kinase domain and C-terminus which are not isotopically labelled. This partly stems from requirement that the kinase domain should be expressed in insect cells to obtain a functional and well-behaved protein. For NMR studies of the kinase domain demands deuteration, which is challenging in eukaryotic expression systems. Another challenge that we faced in this investigation is the complexity and size of the Akt forms as well as their limited stabilities and solubilities for NMR analysis. Despite these impediments, we believe that we have gained important new information about how Akt is regulated by its PH domain in various phosphorylation states and in the presence of allosteric inhibitor. In addition, we posit that these studies will propel additional efforts by our lab and others in future studies to generate an even deeper understanding of this fascinating enzyme.

Comments to address:1) Work by Lucic et al. https://doi.org/10.1073/pnas.1716109115 contains important contributions to understanding AKT regulation and yet is not described or referenced in the submitted manuscript. Please include it in the revised manuscript.

We thank the reviewers for pointing this out. While this work was related to the Ebner et al., 2017 paper, we agree that it is important to cite this study and briefly discuss it. We have now done so in the Introduction and the Discussion of the revised manuscript, noting its use of HD exchange analysis, but pointing out its reliance on the phosphomimic S473D rather than pSer473 that we believe limits its impact on mechanistic interpretations.

2) The authors suggest that the putative regulatory interaction between pSer473 and R144 "exerts tension on the PH-kinase domain linker, thereby pulling on the PH domain and severing its contacts…" The use of the word "tension" throughout is misleading as the authors have not measured force in any of their experiments. Instead, it seems more appropriate to think about the pSer473/R144 interaction as promoting a shift in the conformational ensemble of the full-length protein that leads to destabilization of the PH/kinase interface.

We think that this is a fair criticism and have dialed back the description of tension as the likely basis for the hexaGly insertion. We now mention alterations in tension as a possible basis for the catalytic changes along with changes in length and flexibility conferred by hexaGly insertion in the revised manuscript.

3) The authors claim the results of increasing linker length (addition of the hexaGly insert) "support a role for linker tension," but really linker length has been probed. A simple explanation may be that the increased linker length and flexibility in the region adjacent to R144 might create an increased entropic cost in forming the pSer473/R144 contact. There seems to be an assumption in these experiments that the pSer473/R144 interaction is unchanged upon the insertion of hexaGly.

We fully agree with these astute comments and as noted above, have revised the manuscript accordingly to emphasize linker length and flexibility rather than tension.

4) The NMR data, as presented in Figure 4 are difficult to see. Perhaps the spectra could be expanded to show key spectral changes. More importantly, the NMR changes are never mapped on the PH domain structure. Figure 4F simply shows regions of secondary structure, but the data itself (Figure 4D and E) could be significantly more informative if mapped onto the PH domain structure. This is true for all of the HSQC data throughout the paper.

We agree with the reviewers that it is difficult to appreciate the changes in the HSQC spectra of the original manuscript. To remedy this, we have created two new figures in the revised manuscript. Figure 4 shows an enlarged overlay of the HSQCs that we believe are easier to evaluate. Furthermore, Figure 4C shows a few selected peaks in a “zoomed in” manner. Figure 5 now includes representations of the most significant chemical shift perturbations highlighted on the structure of the MK2206-inhibited Akt relative to the signals of the pSer473 and pSer477/pThr479 Akts (Figure 5D and E). Likewise, Figure 8 now focuses on the NMR data for the effect of MK2206 and includes expanded peaks (Figure 8B) and a representation of the chemical shift perturbations projected on the three-dimensional structure (Figure 8G).

5) What are the phosphorylation states at T308 for all of the protein samples generated via semisynthesis? It appears that Thr308 is phosphorylated in the protein samples prepared for the NMR work. This is puzzling as the authors are probing the transition from autoinhibited to activated AKT via C-tail phosphorylation. So it seems that as long as the activation loop is phosphorylated they won't capture the fully autoinhibited form. Moreover, the scheme in Figure 1B indicates that C-tail phosphorylation occurs before activation loop phosphorylation providing another reason to study C-tail phosphorylation effects in the absence of activation loop phosphorylation. Please clarify and comment on this issue. Can the authors provide MS data confirming the identities (and purity) of all semisynthetic products generated in this study? If additional data supporting the analysis are available, it might be worth including in the manuscript.

In the production of purified Akts analyzed by NMR or enzymatically, the Thr308 has been phosphorylated at near stoichiometric levels by PDK1 either by co-expression or in vitro kinase treatment in all cases but one. This exception was for the study of the DVD to GPG triple mutant that was prepared specifically for kinetic analysis of PDK1 treatment of the full-length semisynthetic Akt in the DVD/GPG context. We performed this experiment since the HCT116 transfection experiments with DVD/GPG Akt unexpectedly revealed diminished Thr308 phosphorylation in response to growth factor. We have clarified this status in the revised manuscript.

In all of the other Akts including the DVD/GPG Akt that was prepared for determining the kcat/Km values of its own kinase activity, Thr308 was phosphorylated in this study. The reason that we elected to do this is based on previous work from our team (Chu et al., 2018) which showed that, in the absence of the PH domain, C-terminal phosphorylation is inconsequential for achieving full kinase activity. In contrast, Thr308 phosphorylation still matters with kinase domain only. In this prior work, both Thr308 and Ser473 phosphorylation in the context of full-length Akt provide about equally important and synergistic impacts on Akt activity. If either one is missing, the kinase activity is similar to having no phosphorylation at both Thr308 and Ser473, although Thr308 phosphorylation confers a lower Km of peptide substrate whereas Ser473 phosphorylation does not.

With respect to the cartoon in Figure 1B showing C-terminal phosphorylation preceding Thr308 phosphorylation, this is by no means established and different labs have concluded an ordered or random process about whether Thr308 or pSer473 comes first. We have added a disclaimer to the figure legend to clarify this point.

It is an interesting question raised by the reviewers regarding how Thr308 phosphorylation would impact the conformation of Akt and agree that this could also be investigated by NMR. Based on our previous measurements that a non-phosphorylated Thr308 form of Akt bound PIP3 about 2.5-fold more tightly compared with pThr308 Akt, there are likely at least subtle differences in the PH domain of the autoinhibited states +/-pThr308. We hope to analyze this issue further using NMR in future studies. We note, though, that the enhanced of affinity of Akt for PIP3 with non-phosphorylated Thr308 Akt is the opposite of what is observed with Akt in complex with allosteric inhibitor and this underscores the distinct Akt conformations that occur with MK2206 vs. MK2206-free Akt.

Regarding protein purity, we have relied on coomassie-stained SDSPAGE rather than mass spectrometry since the early stages of this project. In some unpublished mass spec analysis from our lab on Akt from several years ago when we first started working with the enzyme, we saw evidence for multiple PTMs including additional phosphorylations in the insect expressed component of the semisynthetic protein that was produced. We do not know the stoichiometries of these additional phosphorylations and have not mapped the sites. We do not have a simple way to remove the extra phosphorylations since treatment with alkaline phosphatase would hydrolyze the critical pThr450 that stabilizes Akt. So we have lived with this PTM heterogeneity, and we concede that this can complicate the interpretation of the structural and enzymatic studies. In the revised manuscript, we now allude to this limitation in the Results section, but also point out that within a set of experiments to assess different C-tail modification effects, we use the same batches of recombinant core kinase or PH-kinase to ensure apples to apples comparisons.

6) The structural distinction between the natural auto-inhibited state and the drug-induced state is interesting but poorly defined here. A potentially important finding is that the AKT crystal structures containing small molecules that “staple” the PH domain to the kinase domain may not be revealing the native autoinhibitory interface. The NMR mapping in this paper could provide insight into the PH/kinase interface in the absence of drug, but the authors stop short of using their data in this way. Is there some reason that the data have not been pushed to provide a more definitive understanding of the PH-kinase interaction? It would be illustrative and helpful to see the residues with significant chemical shift perturbations in Figure 6 highlighted on the structure of the drug-bound autoinhibited state. A discussion of how the extended interface upon drug binding differs from the native autoinhibited state is warranted. What can be inferred from the cross saturation transfer experiments? Was such an experiment done in the presence of the allosteric inhibitor? This experiment isn't necessary, but a general discussion of structural models that can be inferred from the data in this manuscript would strengthen the manuscript.

As explained above, Figure 8 now focuses on the NMR data for the effect of MK2206 and includes expanded peaks (Figure 8B) and a representation of the chemical shift perturbations in the context of the Akt structure (Figure 8G). We also performed CST experiment for the non-phosphorylated full-length Akt in the presence of MK2206 (Figure 8B, expanded peaks from CST spectra). One of the challenges we faced with the CST experiments is the sensitivity of the measurement. In addition to concentration limited, the CST experiments were conducted in 70% D_2_O to minimize, water mediated spin diffusion, which further reduced the intensity of the weaker peaks. In response to the reviewer’s comment, we mitigated this problem by considering CST data from groups of neighboring amino acids and we present the CST data in a corroborating manner (Figure 8E and F, CST data as bar plot and statistical bar and whisker plot, respectively).

Interestingly, none of the most affected residues in this context directly contact MK2206 in the crystal structure (this may be partly due to lack of assignment for some residues). From the chemical shift perturbation and CST data, we can infer two main phenomena: (i) the short hinge associated with aa 44-46 is “released” (CSPs are smaller and CST weaker in the presence of MK2206) and (ii) the C-terminal helix is interacts more tightly with the kinase domain and/or C-terminal tail. Additional experiments would be needed to characterize these findings in greater detail, but we speculate that the results suggest one or a mix of two possibilities when MK2206 binds to autoinhibited Akt: i) a reorientation of the PH domain with respect to the kinase domain or C-tail; ii) conformational/dynamic rearrangements within the PH domain itself.

We now include the following lines in the revised manuscript based on the CST data:

“We also performed a CST experiment in the same manner as described above, in the presence of MK2206 (Figure 8B,E and F) and observed two principal effects mediated by the compound. The short hinge of aa 44-46 seems to separate further from the kinase domain and/or C-terminal tail while the C-terminal α-helix, in contrast, appears to interact more tightly with the kinase domain or C-tail.”

We modified the manuscript to clarify the possible interpretation of NMR data:

“These inhibitor-induced changes likely result from a combination of altered structural and dynamic features relative to the baseline autoinhibited form of Akt. These could include a reorientation of the PH domain with respect to the kinase domain or C-tail (placing the C-terminal helix closer) as well as conformational and dynamic rearrangements within the PH domain itself.”

7) Cross-saturation transfer NMR data are also underutilized. Examining the quantitative differences between non-p-C and pS475 should provide more precise insight into how serine phosphorylation might be affecting the conformational preference of the PH domain. It's difficult to see from Figure 4—figure supplement 3 how the cross-saturation transfer effect is stronger for non-phosphorylated versus tail-phosphorylated samples, as stated in the text. Could this be quantified in some way – perhaps showing the mean peak intensity ratios for each sample or a distribution of these ratios across all samples?

We thank the reviewers for this insightful comment. Although we think that interpreting the average of the CST effect over the entire PH domain might not be so edifying, we agree that the CST data can be presented in a more quantitative way (and we have explained the challenges in interpreting the CST data above). We have added a section in Figure 5 of the revised manuscript in which the CST effects over the four parts of the PH domain that we defined are presented for each phospho-state (Figure 5G). Statistical analysis of these PH domain parts revealed that the only significant differences are found for the short hinge (in which pSer473 makes it looser while pSer477/pThr479 makes it tighter) and for the C-terminal helix which shows a tighter interaction only in the pSer477/pThr479 case. We have highlighted these correlations in the revised manuscript.

8) The heavy focus on the loop following b3 (residues Gln43 and Asp44) seems driven (according to the text) by more dramatic chemical shift perturbation and more extensive line broadening in the phosphorylated forms compared to non-p-C. This seems a stretch as there are certainly other regions that show equally or nearly the same extent of spectral change (it's possible that the HSQC data itself makes the differences more evident, but expanded spectral data showing individual HSQC peaks are not provided).

We modified Figure 4 accordingly to add a few selected peaks, including Gln43 and Asp44, in an expanded spectrum (Figure 4C).

We included the following lines in the manuscript based on the CST data:

“Similarly, the CST behavior for Gln43 and Asp44 was closely linked on the phospho-state of the C-tail (Figure 5G and Figure 5—figure supplement 3) with an apparent loosening of the interaction for pSer473 but an apparent tightening for pSer477/pThr479.”

9) The remainder of the paper focuses on a mutation of the native DVD sequence (residues 44-46) to GPG in an effort to "favor a random coil secondary structure in this region." The authors do not characterize this mutant to determine whether the mutation successfully shifted secondary structure preference of the PH domain but instead put the mutation directly into their full-length semi-synthetic AKT. Despite the lack of characterization of the mutant, this is the one place in this paper where the authors see a dramatic result; Figure 5A shows that the mutation of the DVD motif causes a large decrease in the steady-state kinetics of AKT in its pSer473 form compared to wild type. The other phosphorylated form of AKT (pSer477/pThr497) is unaffected by the mutation in the PH domain. Clearly understanding the mechanistic basis for this difference would be an advance in this field but the manuscript as presented does not sufficiently explain the observations.

We agree that a detailed understanding of the DVD hinge in Akt regulation is incomplete. We think that the CST effects that the reviewers encouraged us to more quantitatively analyze, nicely correlate with the divergent catalytic impacts of DVD mutation between the two C-tail phospho-states. We believe that this unexpected structure-activity relationship in this part of the PH domain between the pSer473 and pSer477/pThr479 forms adds to our overall understanding of how Akt regulation is achieved by distinct patterns of C-tail phosphorylation.

10) In the Discussion, the authors interpret the failure of pSer473 to stimulate AKT activity as being due to changes in the "normal dynamics" of the DVD motif. Dynamics can be directly measured by NMR to support this conclusion, but this is surprisingly not included in the manuscript. The observation that activation loop phosphorylation (at Thr308) is also diminished upon mutation of the DVD motif to GPG might suggest that the mutation in the PH domain enhances the autoinhibitory contact between PH and kinase domains and/or alters the accessibility of the activation loop to PDK-1. Unfortunately, as presented, the manuscript provides little to advance our understanding of the precise role of this region of the PH domain in AKT regulation.

We agree with the reviewers that NMR dynamics experiments would be the best way to validate this hypothesis. However, typically a set of relaxation, CEST or relaxation-dispersion spectra takes about 40-60 times longer than a single 2D (due to their pseudo-3D nature and the longer recycling delays they require). In our case we are limited by concentration and sample stability. In our case a single 2D ^15^N-TROSY-HSQC took about 50 hrs and at the end of the experiment, we saw heavy precipitation, which makes such dynamics experiments infeasible. In the revised manuscript, we modified the text to make clear that our hypothesis about the dynamics is inferred from the previously published crystal structures rather than from NMR data:

“This suggests that the normal dynamics (loop to helix transition) of this DVD motif is necessary for allowing the PH domain to dislodge from the kinase domain in response to pSer473.”

11) In the final part of this work the authors explore the effects of drug binding on PIP3 binding as well as PH domain NMR chemical shifts. The data are not interpreted in a satisfactory manner; the authors state that "Notably, some of the most significant chemical shift perturbations were detected in the PIP3-binding interface…" but there are significant changes elsewhere in the PH domain as well. There is also no attempt to take into account the likely significant spectral changes that will arise from the drug itself.

We now recognize this shortcoming. As discussed above, we have discussed the MK2206 NMR changes in greater detail in the revised manuscript.

12) Finally, the last sentence of the paper suggests conformational heterogeneity might be present in the non-phosphorylated C-tail form of AKT. This is very logical and raises the question of whether the authors see any evidence for the existence of multiple states in solution for any of the segmentally labeled forms they have studied in this submitted paper or the previous 2018 publication?

We did not detect any proof of conformational heterogeneity in our spectra (e.g. peak doubling in the HSQCs). However, this can be due to a number of limitations including insufficient signal-to-noise ratio (due to low stability and long measurement times required) and incompatible time regimes (fast to intermediate exchange). We would have loved to perform relaxation dispersion or CEST experiments, which can shed light on the presence of minor states. As explained above, NMR dynamic experiments that theoretically are able to characterize such phenomenon are inaccessible in this experimental system. Therefore, we cannot address such heterogeneity by NMR at this stage. We have added the following sentence to the revised manuscript:

"Although NMR dynamics experiments due to their very long time scales are not feasible with our current experimental system, future alternative approaches may allow this to be investigated."

13) Some of the kinetic parameters for the WT semi-synthetic AKT proteins shown in Figure 2 seem to differ a little from the parameters reported earlier in Chu et al., 2018. For example, the kcat/Km for A2 (full-length AKT phosphorylated on T308) in Figure 2 of Chu et al. is 28-fold smaller than the value reported in the current manuscript. In the present manuscript, the concentration range of ATP used is 0-2000 µM, whereas the Km values reported for A2, A1, and A6 (Figure 2 of Chu et al.) are 200, 5100, and 3700 µM, respectively. Perhaps the authors could clarify the apparent discrepancy.

The reviewers are correct that the catalytic parameters of the wt phospho- and non-phospho-Akt enzymes analyzed here do not precisely match the data in Chu et al., 2018, they are within 3-fold for each state (and 2.8-fold but not 28-fold different for A2). We speculate that this may be due to some heterogeneity of PTMs in the recombinant portion of the enzymes made in insect cells and have added this potential explanation to the revised manuscript.

14) Perhaps mention that the reported kcat/Km's are apparent values as the second substrate (the peptide in this case) is not present at a saturating concentration.

We have added the qualifier "apparent" as suggested by the reviewers. However, we previously showed that the Km of the GSK3 peptide for all forms of Akt analyzed that contain pThr308 are ca. 3 μm meaning that we are likely working with saturating peptide substrate concentration in the cases analyzed here.

15) It would be helpful in Figure 1 if the various states were labeled as "active" or "inactive." In Figure 1B, the ribbon diagram of the PH-kinase unit is drawn upside. In this and other figures (e.g., 4F), it would be helpful if structural components and parts were clearly labeled on the figures.

In the revised manuscript, we modified all of the figures displaying the Akt structure to show the canonical kinase domain structure with the N-lobe on top (e.g. Figure 1B, Figure 5D,E,F, Figure 8G). We also labeled the different domains of Akt as well as the allosteric inhibitor MK2206 in Figure 5F (previous Figure 4F).

16) Can the authors comment on the purity of the samples used in Figure 2? Based on the figure supplement SDS-PAGE analysis, there appears to be a ~10-20% impurity. Can this be identified by MS analysis of the full-length protein to confirm that it isn't a species that will interfere with the analysis of enzyme kinetics?

The semisynthetic protein samples displayed in the SDSPAGE Figure 2—figure supplement 1 show about a 10% impurity. This impurity likely results from incomplete ligation of the C-tail peptide and/or trace proteolysis at the N- or C-terminus. Since prior work from our lab and others indicate that C-tail deletion of Akt is severely catalytically impaired. Moreover, these Akt forms have a 40 aa epitope tag at the N-terminus and we have shown previously (Chu et al., 2018) that the this tag (and its removal) has little or no effect on catalytic activity. Thus, we do not believe this impurity band should impact the interpretation of mutations/phosphorylation and their influence on kcat/Km values of the Akt forms in this study.